# Development of forecast information for institutional decision-makers: landslides in India and cyclones in Mozambique

Mirianna Budimir[1], Alison Sneddon[1], Issy Nelder[1], Sarah Brown[1], Amy Donovan[2], and Linda Speight[3]

[1] Practical Action Consulting, Rugby, CV21 2SD, United Kingdom
[2] Department of Geography, University of Cambridge, Cambridge, CB2 3EN, United Kingdom
[3] University of Reading, Reading, RG6 6UR, United Kingdom

*Correspondence to*: Mirianna Budimir (Mirianna.budimir@practicalaction.org.uk)

**Abstract.** There remains a gap between the production of scientifically robust forecasts, and the translation of these forecasts into useful information such as daily "bulletins" for decision-makers in early warning systems. There is significant published literature on best practice to communicate risk information, but very little to guide and provide advice on the process of how these bulletins have been, or should be, developed. This paper reviews two case studies where bulletins were developed for national and district-level government agencies and humanitarian responders: daily reports in response to Cyclones Idai and Kenneth in Mozambique, and prototype landslide forecast bulletins in Nilgiris and Darjeeling Districts of India. Primary data was collected from producers and intermediaries of the bulletins via interview, and secondary data analysed on: iterative changes in the bulletin development; minutes from internal discussions; and feedback from users to extract learning on both the content and process of developing the bulletins. There were significant similarities in the type of content included in the bulletins, such as the layout, choice of words, and use of visualisation that was consistent with published best practices. Both case studies experienced challenges dealing with uncertainty, complexity, and whether to include advice. There were also similarities in the processes and approaches taken to develop the bulletins. Both case studies took an iterative approach, developed feedback mechanisms, benefitted from experienced multi-disciplinary teams, emphasised the need for strong inter-relationships, and the importance and value of preparedness and protocols. A major challenge was the difficulty of balancing science capabilities, including issues related to data scarcity, with user needs, which did not become significantly easier to deal with given more time availability. In particular, there were tensions between developing new forecast products that were urgently needed by users, against the limited time for testing and refinement of those forecasts, and the risk of misinforming decisions due to uncertainty of the information based on limited data. The findings indicate that whilst more research is needed into existing or best practice processes to develop content for forecast bulletins, there is an existing body of experiential and intuitive knowledge and learning that already exists but is not yet captured in an appropriate format that could be of significant interest and value to those developing forecast information. This paper goes some way to capturing some of the learning from translating scientific forecasts into useful information, in particular on both the content and the process of developing forecast bulletins for decision-making.

# 1 Introduction

There remains a gap between the production of complex scientific forecasts and warnings and the operational use of such information by institutional stakeholders in official decision-making roles such as government officials or civil society actors operating in a preparedness, risk reduction, or response capacity, who have a wide range of educational and professional backgrounds and information needs (Morss et al., 2005; McInerny et al., 2014; Stephens et al., 2015; Cumiskey et al., 2019). In particular, there are limited evidence-based guidance publications that document the process of developing natural hazard-related forecast products for institutional decision-makers, and therefore limited opportunity to learn from the experience of others. This gap is beginning to be filled by recent publications from forecast and early warning programmes and initiatives that focus on research-into-action such as Hammersmith et al. (2020) and Zhang et al. (2019). Continuing to close this gap for effective action and informed decision-making is an important priority for scientists within the hazard community, if we are to achieve impact.

As part of the Foreign, Commonwealth and Development Office (FCDO) and Natural Environmental Research Council (NERC) funded Science for Humanitarian Emergencies And Resilience (SHEAR) programme, two case studies have emerged on the development of early warning and forecast information for institutional decision-makers, for different hazards, and with different time pressures on the development of bulletin information. The case studies developed forecast information for national and district-level government agencies and humanitarian responders: daily reports in response to Cyclones Idai and Kenneth in Mozambique, and prototype landslide forecast bulletins in Nilgiris and Darjeeling Districts of India.

These case studies provide an opportunity to learn from these experiences, and extract relevant knowledge to guide others in the development and production of forecast products for institutional decision-makers. They provide an interesting opportunity to learn about the process of developing early warning and forecast information in the form of regular reports or "bulletins" for institutional decision-makers: what information and content should be included; how should it be framed; who needs to be involved in the development of the information, and what skills or perspectives do they bring to the process; what are good and bad practices of developing bulletin information; and what can others learn from the experiences of those involved?

Throughout this paper, the term "bulletin" will refer to the forecast information product produced by either team. Bulletins are commonly used across the world by National Hydro-Meteorological Services and disaster risk managers to provide regular (typically daily or weekly) forecast information related to weather, hazards, and potential impacts specifically for institutional decision-making stakeholders, such as government officials and civil society actors. In this context, the cyclone forecast bulletins contained information on weather, hazard and potential impacts and were designed to be used alongside local information to support humanitarian decision-making for example where to set up emergency shelters; the landslide forecast bulletins contained information on forecasted landslide probability and existing landslide susceptibility and were designed as part of an experimental prototype forecasting system. In the landslide case study, the bulletins were a two-page product; in the cyclone case study, they were an 8-15 page document referred to internally within the team as a "report".

Within this paper, "producers" will refer to those physical or forecast scientists who developed the forecast information and produced the report. "Users" will refer to those organisations that received the forecast bulletins. "Intermediaries" will refer to organisations that act as a knowledge and/or relationship link between producers and users with the aim of developing applied, practical, scientifically robust, and useful information (Figure 1).

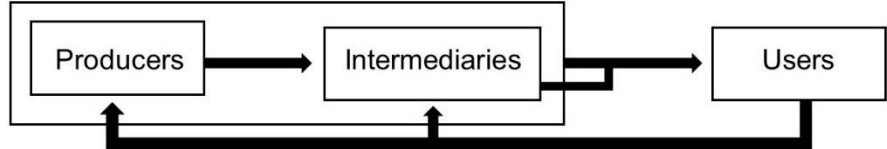

**Figure 1 Diagram indicating common relationship patterns between the roles of producers, intermediaries and users in bulletin development and production. Arrows indicate typical flows of information.**

This paper will cover: a review of the literature on bulletin content and development and the importance of trust; provide background information to the landslide and cyclone case studies; outline the methodology and data sources used for analysis in this paper; describe results; discuss key points of interest; and summarise conclusions.

## 2 Literature Review

The following section summarises key findings within literature from across a variety of disciplines. The first section is intended to highlight consensus within literature regarding the processes needed to develop these products. There is a noticeable absence of any publication which summarises these processes in setting up and developing these types of products, particularly for developing natural hazard-related forecast 'bulletins', however a number of transferable best practices and learning from the development and use of similar forecast products, risk information, and climate change communication have been identified and discussed below. The second part will discuss the different components of forecast products, and the aspects that should be considered and understood in order to develop the most appropriate product. The third section emphasises the role of trust in developing and creating content for useful forecast products.

### 2.1 Bulletin development

Whilst there is existing guidance on what to include in risk information and there is wider related literature that can provide some guidance and advice on this development process to create forecast bulletins (e.g. working in an inter-, multi-, or trans-disciplinary way), there does not appear to be any literature specifically providing guidance on the process of developing forecast bulletins for the user groups defined in this study, i.e. professional, institutional user groups (Stephens et al., 2015). This section summarises some of the general recommendations pertinent to developing forecast bulletins.

Trans-disciplinary collaboration is important in developing forecast products, bringing in diverse expertise, knowledge and perspectives (Morss et al., 2005; McBean and Rodgers, 2009; McInerny et al., 2014; Anderson-Berry et al., 2018; Robbins et al., 2019; Lemos et al., 2012). For example, meteorologists working on bulletins need to go beyond a narrow role, collaborating across disciplines (McBean and Rodgers, 2009). Cumiskey et al. (2019) highlighted that each individual involved in the creation of a forecast product brings with them a unique set of experiences and skills, and that effective multidisciplinary co-development requires care and planning to ensure each expert makes recommendations and provides guidance (only) on their area of expertise. Morss et al. (2005) emphasise the importance of defined and agreed roles and responsibilities.

Identification of bulletin users and equitable co-development with or tailoring to that audience is an integral part of an effective forecast product (Harrowsmith et al., 2020; Harold et al., 2017; Carter et al., 2019; Lemos et al., 2012; Vincent et al., 2020). Early warning systems have too often been misconstrued as one-directional methods of communication, where information travels directly from the scientific producers to the target users (Sukhwani et al., 2019; Lemos et al., 2012). A multitude of factors make the development and communication of understandable and actionable forecast information incredibly complex, with complexity in the hazards themselves, alongside complex social, political and economic contexts (Patt and Gwata, 2002). Production of actionable forecasts necessitates understanding of the contexts in which this information is being shared and used (Harrowsmith et al., 2020). Forecast producers need to understand the "...real, on-the-ground, needs of end-users…" to ensure those users will be able to understand the information and make informed decisions based on it (WMO, 2012, p. 8). Recipients of forecast information have diverse needs and preferences in regard to what format they most engage with and understand (Robbins et al., 2019; Harrowsmith et al., 2020; Patt and Gwata, 2002).

Lambrecht et al. (2019) recommends that forecast producers should undertake research to understand the communities they work in, to improve effectiveness and impact of the forecast product (Harold et al., 2017). McInerny et al. (2014) stress the importance of conducting targeted user research early on in the process to ensure products are relevant, understandable and actionable. Robbins et al. (2019) explains that in order for forecasts to elicit the intended response, there needs to be regular "collaborative dialogue platforms" which require sufficient time to build trust and partnerships, proper funding and operating procedures to be successful, as well as mechanisms to support equitable partnerships (Lemos et al., 2012; Vincent et al., 2020; Carter et al., 2019).

An effective forecast product requires long-term equitable partnerships between scientists, users/decision-makers and practitioners (Morss et al., 2005; Harold et al., 2019; Lemos et al., 2012; Vincent et al., 2020). Carter et al. (2019) outline a series of building blocks for co-production of weather and climate services including identifying key actors and building relationships, building common ground, co-exploring needs, co-developing solutions, co-delivering solutions, and evaluation. Bulletin development should be iterative and involve a number of adaptations both in terms of the product and those involved (Morss et al., 2005; Kox et al., 2018; Robbins et al., 2019; Harrowsmith et al., 2020). Forecast information needs to be continuously adapted on account of ever-evolving environments, technologies and cultures, and therefore an iterative process must be maintained (Harrowsmith et al., 2020; Lemos et al., 2012).

A major barrier to effective forecast product dissemination is a lack of understanding of who are key users, a barrier that can be overcome if products are collaboratively co-developed with users (McInerny et al., 2014; Gough, 2017; Kox et al., 2018; Taylor et al., 2018; Robbins et al., 2019). The role of the users should shift from detached user to collaborator and partner, with products developed in a participatory and transparent way (Kox et al., 2018; Lemos et al., 2012) and with users engaged from the beginning of the process (Speight et al., 2018; 2021). Co-production with users improves product quality and usability, as well as helping manage user expectations of the scientific community (Robbins et al., 2019; Patt and Gwata, 2002). Wachinger et al. (2013) found that when communities are involved in designing and testing emergency plans, they are more motivated to listen and take action on information provided during a real event. Robbins et al. (2019) found that perceived unreliability of the information source impeded use, whilst relationships between producers and users of forecast information enabled uptake (Lemos et al., 2012). There is an extensive amount of literature highlighting the need to identify the users to ascertain what their needs are and when they need information, and whilst many studies recommend a collaborative approach, there is little to direct and explain the formal process that would allow this specifically for forecast bulletins. Gill et al. (2008) emphasises the need for an established mechanism with formal channels of communication, but does not detail how such mechanisms and processes could be developed, a literature gap that merits further analysis.

## 2.2 Bulletin content

Harrowsmith et al. (2020) suggest the following information content should be included in forecast products, to align with user needs: what is going to happen, when it will happen, how bad it will be and where, and instructions or guidance to further resources on what can be done to reduce the impacts.

Gill et al. (2008, p. 2) emphasise that "...unless the forecast information is communicated effectively to users, its full value will not be utilised". Forecast information is best communicated through the use of "accessible" (Anderson-Berry et al., 2018) or "plain" language (Robbins et al, 2019), reducing user confusion and enabling better decision-making (Harold et al., 2017). It is still beneficial to include some technical language, for example, in order to communicate uncertainty, but these instances should be accompanied by clarification of the meaning of such terminology in a clear and understandable way (Patt and Schrag, 2003; Harold et al., 2017).

The format and presentation of information critically influences the extent to which that information is understood. The use of images increases user risk perception (Bica et al., 2019; Anderson-Berry et al., 2018; Gough, 2017), understanding (Harold et al., 2019), and risk aversion (Visschers et al., 2009). Visualisations and graphics are found to be the most engaging, especially in settings with multiple languages, where visualisations can reach non-expert audiences across language barriers (McInerny et al., 2014; Harold et al., 2017). Robbins et al. (2019) found maps or images more useful than text alone, especially when conveying technical information which may easily be misunderstood by the non-scientific community. Imagery can reduce the length of forecast products, enabling rapid review by decision-makers (Gil et al., 2008).

The use of visual imagery to communicate risk needs to be carefully considered (Harrowsmith et al., 2020). A known limitation
of visual risk information is the possibility of confusion and misunderstanding. For example, Taylor et al. (2018) showed that
the use of storm polygons leads individuals to misunderstand risk variability within a given area. Accompanying text reduces
the level of misinterpretation, though without eliminating it entirely (Gough, 2017; Taylor et al., 2018; Harold et al., 2019).
Kox et al. (2018) recommend a balance between compactness of forecast products and detail, combining text with images in
a form that can be understood and acted upon (Stephens et al., 2015; Harold et al., 2017). Harold et al. (2017) recommend the
following process for developing effective visuals: identify the main message, assess users' prior knowledge and thought
process, choose visual formats familiar to users, reduce complexity where possible, build up information to provide visual
structure, integrate text, avoid jargon, use cognitive design principles, and test visuals to check comprehension with users.
Preferences for visual formats varies by users and by context, often influenced by factors including culture and educational or
training background (Harold et al., 2017; Fleming et al., 2005).

Communicating forecast uncertainty is discussed extensively in the literature, with consensus on the benefits of including this
uncertainty information (Morss et al., 2005; Harrowsmith et al., 2020). Benefits include user expectation management (Gill et
al., 2008); maintaining trust when forecasts are inaccurate (Taylor et al., 2018); and aiding decision-making (Frick and Hegg.,
2011; Anderson-Berry et al., 2018; Taylor et al., 2018; Bica et al., 2019; Harold et al., 2019). The way in which uncertainty
information is understood is highly dependent on user background and experience and therefore defining the intended user of
forecast information is critical to ensuring the output is appropriate for user understanding (McInerny et al., 2014). Whilst
communicating uncertainty can be difficult, and the language used may differ between scientists and non-technical users
(Lambrecht et al., 2019), the provision of uncertainty information can improve trust in forecasts, and help with decision-
making by users.

Several studies highlight the difficulties the public find in interpreting and understanding uncertainty (Patt and Schrag., 2003;
Budescu et al., 2014; Bica et al., 2019). Stephens et al. (2019) and Nadav-Greenberg and Joslyn (2009) however, found that
when decisions have real-life consequences, the general public can make effective and informed decisions based on uncertainty
information. Their study concluded that the use of uncertainty information is advantageous and results in improved decision-
making and that describing "worst-case scenarios" can be detrimental and should be avoided.

Frick and Hegg (2011) found that the general population seems to favour probabilistic information when it is accompanied by
additional information about uncertainty. The inclusion of uncertainty information increases the public's trust and confidence
in the forecasts, and can help with combating the damaging effect of false alarms (Taylor et al., 2018; Zhang et al., 2019;
Harrowsmith et al., 2020). However, how and what to include in terms of uncertainty information needs to be nuanced to
specific users to avoid confusion (Robbins et al., 2019).

## 2.3 Trust

A lack of trust in producers of forecast or warning information by those who receive and use them to make decisions can be one of the most significant factors affecting user risk perception, and their subsequent actions (Taylor et al., 2018; Harrowsmith et al., 2020; Patt and Gwata, 2002). Anderson-Berry et al. (2018, p. 21) emphasise that "…in the successful dissemination of warnings, it cannot be assumed that warnings will be recognised as such and understood and trusted by recipients…". Trust can be influenced by diverse factors including: previous personal experience with inconsistent forecasts and the relationship

with the source of information (Wachinger et al., 2013; Anderson-Berry et al., 2018; Taylor et al., 2018; Robbins et al., 2019; Patt and Gwata, 2002; Lemos et al., 2012). There are many ways of cultivating trust, including through the content provided, such as the language used and whether uncertainty information is conveyed (Samaddar et al., 2012; Lambrecht et al., 2019; Zhang et al., 2019; Harold et al., 2019). As discussed in section 2.1, another way of improving this sense of trust is to involve the user in the design and testing of plans and products; this can also increase the general public's understanding of what the

science can realistically do, and what the information means (Wachinger et al., 2013; Speight et al., 2018). Trust in the scientific forecasts themselves in terms of accuracy of predictions is also vitally important; evaluating, understanding, and communicating forecast skill transparently can support this (Harrowsmith et al., 2020, Patt and Gwata, 2002; Carter et al., 2019).

## 3 Background

This paper aims to extract key learning and recommendations from two case studies, one focused on landslides in India, and the other focused on cyclones in Mozambique. Both case studies independently developed forecast information products in the form of "bulletins" or "reports" that were produced on a daily or almost-daily basis to provide information on the likelihood of a hazard or its impacts in advance. Users to receive the forecast bulletins were selected based on their existing disaster risk management and preparedness roles within the context.

The two case studies differ in several significant ways. The landslide bulletin evolved over a much longer time period (18 months), during a pre-operational experimental phase of a project, whilst the team enhanced the data and scientific models underpinning the forecasts (however, it should be noted that landslides did occur during the production period of the bulletin), and recipients of the bulletin were instructed not to use the information to inform their actions until the forecast skill could be evaluated. In contrast, the cyclone bulletins evolved during the context of a discrete ongoing humanitarian response over a

period of days-weeks (Emerton et al., 2020) and were actively employed by users.

It should also be noted that the scope of work was very different for each case study – the cyclone case study was building mostly on previously developed forecast models and output, whilst the landslide case study was developing a completely new forecast product. The landslide project needed to manage a wide number of areas of new science, development of new datasets and validation of new approaches to landslide forecasting in South Asia, a significant task that requires many years of data

collection, model testing, and refinement. There was pressure to move towards working on system operationalisation when the models, datasets and science underpinning the forecasts were still at an early stage.

The case studies had differing numbers and categories of users, with the landslide bulletin targeting a very small number of sub-national government users, whilst the cyclone bulletins were shared more broadly. The case studies also differed in the number of producers and intermediaries involved. The landslide bulletin involved a bigger number of producers with the project bringing together scientists from different disciplines as well as building capacity and sharing knowledge across producers from three countries. The cyclone bulletin development took place in a tight and urgent time frame with a smaller operational team.

The data used for this paper covers a specific period within the development of these bulletins, but there was continued evolution of the bulletins beyond this study. This paper focuses on what the process was during this period, reflections on the process of co-creating bulletins, and the evolution of bulletin content during this timeframe. An analysis of the use of the bulletins by users is beyond the scope of this project (as users were not interviewed and the landslide project was operating as an experimental prototype system, with instructions given to users not to actively use the forecast information for decisions), but would be a valuable addition to the global body of knowledge on effective practice in this field. This topic is addressed to some extent in Emerton et al. (2020). This paper also does not aim to explore the piloting and operationalisation of new risk products, and does not review practical and ethical issues of trialling new risk products amongst at-risk populations. This is noted as a limitation, and an area for further research.

### 3.1 Landslide early warning bulletins in India

The SHEAR LANDSLIP (Landslide multi-hazard risk assessment, preparedness and early warning in South Asia integrating meteorology, landscape and society) project (2016-2022) is working in India to develop a prototype rainfall-induced landslide early warning system at district scale, piloting it in two study sites in Nilgiris and Darjeeling Districts. Within LANDSLIP a daily bulletin for the monsoon period is being collaboratively and iteratively developed by a range of technical and specialist project partners for experimental use. The users of the bulletin in this case study are sub-national government (District Authorities in Nilgiris and Darjeeling).

The first version of the bulletin was developed in November 2018 and underwent multiple iterations over the course of 18 months. The bulletin was generated on a daily basis during the 2019 monsoon to test procedures and evaluate forecast skill, but was not sent "live" to the users. The bulletins were again generated during the 2020 monsoon period and this time sent to sub-national government officials with instructions not to share the information further, nor to use the forecast information to make decisions during this period where the forecasts were under evaluation.

This paper focuses on the period of bulletin development between November 2018 and July 2020, but it should be noted that the bulletin has undergone further revisions since July 2020. Although these developments are not discussed in this paper, the

learning outlined in this paper and the knowledge generated through the project teams' experience of developing the bulletin has been taken forwards and work is ongoing with Geological Survey of India to improve the production and use of landslide forecast bulletins in support of a soon to be established National Landslide Forecasting Centre.

**3.2 Cyclone Idai and Kenneth forecast reports for Mozambique**

The SHEAR FATHUM (Forecasts for Anticipatory Humanitarian action) project is working in Mozambique, Uganda and South Africa to improve the use of forecasts to support taking early action in advance of a flood occurring. Cyclone Idai was named as a tropical cyclone on 12th March 2019. It made landfall near Beira, Mozambique on 15th March. After landfall it quickly dissipated but continued to bring continuous rainfall for several days leading to widespread flooding in central Mozambique (Emerton et al., 2020). The forecast uses the Global Flood Awareness System (GloFAS, www.globalfloods.eu),

an early warning component of the European Commission Copernicus Emergency Management Service (emergency.copernicus.eu). Based on FCDO's knowledge of FATHUM's work on flood forecasting for humanitarian aid, and their understanding of the high and prolonged risk of flooding following a cyclone's landfall, they requested FATHUM researchers to work with the European Centre for Medium-Range Weather Forecasts (ECMWF) to produce daily hydrological forecast reports in response to Cyclone Idai (Emerton et al., 2020). Recognising a need to incorporate as assessment of potential

impacts, the SHEAR HYFLOOD (Next generation flood hazard mapping for the African Continent at hyper-resolution) project at the University of Bristol was approached to produce flood maps and impact forecasts, estimating the population exposed to potential flooding. The team began producing almost-daily reports on weekdays from 22nd March until 1st April 2019. The first integrated flood hazard and exposure report was produced on 25th March.

The success of the process led to FCDO approaching the same team the following month before Cyclone Kenneth was forecast

to make landfall in Mozambique to produce daily reports in support of that response. Cyclone Kenneth developed as a named tropical cyclone on 24th April 2019, reached peak intensity and made landfall on the evening of 25th April, and dissipated by 29th April. Reports were produced by the cyclone bulletin team on 24th April, 29th April, 1st May and 3rd May. The frequency of report production for Idai and Kenneth was based on user need, availability of new information (either an updated forecast showing significant changes or new observations from the ground), and team availability (weekdays only).

Following the bulletin production for Cyclones Idai and Kenneth, the University of Reading, ECMWF and University of Bristol team has been formally contracted by FCDO (alongside HR Wallingford and FATHOM) in a pilot project to develop flood early warnings. Forecast bulletins have since been produced for Hurricane Iota in Central America (November 2020) and Tropical Cyclone Eloise in Mozambique (January 2021). Although these recent events are not discussed here, the learning outlined in this paper has been taken forwards and work is ongoing with FCDO to improve the production and use of flood

bulletins in support of humanitarian action (for example, a standard terminology guide and event review protocol are being developed).

## 4 Data and methodology

This study draws upon primary data from key informant interviews (KIIs), and secondary data from meetings, workshops, focus group discussions, internal communications, and iterations of the bulletins themselves.

Semi-structured interviews were conducted with 18 key informants, seven involved in developing the cyclone bulletins, and 11 involved in developing the landslide bulletins. At least one representative from each organisation involved in producing or acting as intermediaries for the bulletin were interviewed. The interviews were conducted between August and September 2019 and were framed around their experiences so far in developing the bulletins, identifying challenges, how the bulletin changed over time, how those decisions were made, and what they had learned during the process. All participants were asked 285 for their consent to participate in the research. The interview recordings were transcribed, pseudonymised and handled under the General Data Protection Regulation 2018. An ethical review was carried out for this research at King's College London under the SHEAR programme.

Interview data was combined with three other secondary data sources: written feedback submitted from users to producers whilst the bulletins were being refined; copies of the iterations of the bulletins; and minutes and notes from internal project 290 discussions and meetings. These secondary data were analysed for changes to the bulletin and compared with discussion points and decisions made around content.

Research data was qualitatively analysed in NVivo using a two-stage coding process. Interviews were first coded against key themes identified in the literature review, whilst also considering emergent themes beyond those apparent in the literature - consistent with a constructivist grounded theory approach (Charmaz, 2006). The coded data was then reviewed to identify 295 convergent and divergent themes between the case studies (landslide and cyclone bulletins).

The choice of the case studies was based on the authors involvement within the SHEAR programme. The authors of this paper have occupied various roles within the SHEAR programme including: consortium members in the LANDSLIP and FATHUM projects; team members involved in the development of the bulletins; and/or those acting as Knowledge Brokers of the SHEAR programme. In the process of carrying out these roles, the authors witnessed challenges, and commonalities and differences 300 between approaches and solutions for each case study and identified these examples as presenting an opportunity for learning about the process of developing bulletins from those who were involved.

The authors of this paper bring a range of roles and a unique dual perspective to these case studies, bringing together perspectives from academic and practitioner positions, and core team members of both case studies (bringing an insider perspective), alongside those outside of the core projects who have engaged with those initiatives and teams over several years 305 as Knowledge Brokers of the wider SHEAR programme (bringing a semi-outsider perspective). The authors have made efforts to focus reporting of the results directly from the data sources, ensuring all perspectives are represented, whilst also reflecting on useful learning during the discussion section, to bring in their unique position and experiential knowledge.

## 5 Results

Findings and recommendations distilled from the interview and case study analysis are here presented in nine thematic areas: iterative development and the role of co-production; content including layout, text, visuals, and the inclusion of exposure, vulnerability and impact information; communicating complexity; team roles and skills; priorities and relationships; accuracy and evaluation; understanding users; and preparedness and protocols.

### 5.1 Iterative development and the role of co-production

A clear common finding from both case studies was the iterative process taken to develop the forecast information. In both cases, irrespective of time constraints, hazard type, user groups, or location, the forecast report/bulletin went through several phases of development and multiple changes were made in response to feedback received from users and intermediaries, and discussion within the producer teams (see Table 1 and 2). The landslide bulletin iterations were spread out over an 18-month period while the project team worked on advancing the underlying science and datasets in a novel application in an area with limited data. The cyclone bulletin iterations occurred in a very short window, while the bulletins were in active use for humanitarian response.

**Table 1 Timeline and sources of feedback for the landslide bulletin development**

| Date | Bulletin development/feedback |
|---|---|
| November 2018 | First draft of the bulletin created by producers. |
| November 2019 – January 2019 | Internal discussions between producer and intermediary team members. |
| January 2019 | Feedback received from key producer team member. |
| February 2019 | Internal discussion between producers and intermediaries at in-person project meeting. Draft example bulletin shared with users. |
| June 2019 | Week-long workshop between producers and intermediaries focused primarily on the bulletin. |
| July-Sept 2019 | Feedback from producers and intermediaries over summer after trialing daily bulletin (not shared with users). |
| Sept-Nov 2019 | Multiple internal project video calls to discuss between producers and intermediaries. |

| | |
|---|---|
| Nov 2019 | Week-long workshop between producers and intermediaries focused primarily on the bulletin. |
| November-December 2019 | Bulletin template shared and discussed with users in study sites. |
| February 2020 | Internal discussion between producers and intermediaries at in-person project meeting. |
| June 2020 | Producer's internal discussion, approval and sign off. |
| 25th June 2020 | Project team call to agree content before sharing with users for experimental, closed trial. |
| 1st July 2020 | Bulletin issued daily to users in experimental, closed trial. |

**Table 2 Timeline and sources of feedback for the cyclone bulletin development. Note that between each report issued, feedback and edits were made between intermediaries and producers.**

| Date | Event timeline | Bulletin development/feedback |
|---|---|---|
| 15th March 2019 | Cyclone Idai makes landfall. River levels start to rise. | |
| 19th March 2019 | President of Mozambique declared a state of emergency and requested international assistance. | |
| 20th March 2019 | Slowly falling river levels from this point. | Producers requested to provide reports on flooding from Cyclone Idai. |
| 21st March 2019 | Cyclone Idai dissipates. | Cyclone Idai briefing activity begins with a joint phone call between producers and intermediaries. |
| 22nd March | | First report issued as separate documents for flood hazard and exposure (six Cyclone Idai reports issued during the period 21st March – 1st April). |
| 25th March 2019 | | Flood hazard and exposure information integrated into a single report. |

| | | |
|---|---|---|
| 1st April 2019 | | Final Cyclone Idai daily report issued. |
| 12th April 2019 | | Debrief call between producers and intermediaries on Cyclone Idai. |
| 23rd April 2019 | Kenneth named as a tropical storm | United Nations Office for the Coordination of Humanitarian Affairs (UN OCHA) request reactivation of the flood bulletins via intermediary |
| 24th April 2019 | Kenneth upgraded to tropical cyclone | Cyclone Kenneth briefing activity begins. First report issued (four reports issued during the period 24th April – 3rd May). |
| 25th April 2019 | Cyclone Kenneth makes landfall in Mozambique. | |
| 26th – 28th April 2019 | Localised flooding begins. Significant rise in river levels from 28th April for all major rivers in the region. | |
| 29th April 2019 | Cyclone Kenneth dissipates | |
| 3rd May 2019 | Water levels fall back below alert level | Last daily report on Cyclone Kenneth issued |
| 24th June 2019 | | In-person debrief from Cyclone Kenneth between producers and intermediaries. |
| 20th September 2019 | | Workshop in Mozambique between producers and users. |

In both case studies, bulletins evolved through a number of iterations (Figure 2 and Figure 3), with decisions including, excluding or adapting certain types of content or information, in response to feedback that information was not valuable, not well-enough understood by users, not clear, beyond the scope of the project or team capacities, or potentially dangerous or misleading to include (see section 5.2.4 on exposure, vulnerability and impact information). Feedback loops between
330 producers, intermediaries and users shaped understanding of what information was understandable, relevant and useful for

informing user action. Discussions centred on the bulletin development also helped to shape and inform users' understanding of the scientific capacities of the forecasts themselves.

> "The key thing for me is that these reports did actually grow and change quite a lot." – Cyclone project interviewee 5.

"Normally we had, at the very beginning, maybe three iterations I think, for the first bulletin, between [the intermediary] and the scientists. At the end I think just one iteration was necessary. We had learned." – Cyclone project interviewee 3.

The bulletins produced for Cyclones Idai and Kenneth were intended to inform and support decision-making in the midst of a

340 humanitarian crisis, and therefore time pressure was reported as the most significant challenge facing the teams. The bulletins were issued on a daily timeframe, with the teams working to interpret forecasts and produce a draft for intermediaries to review and provide feedback, then incorporating that feedback and submitting a revised bulletin for circulation by the end of the day. In instances where forecast data was not available until later in the morning, for example, the turnaround time was affected. In general, interviewees found that the time pressure prevented the teams from being able to fully explore effective and useful

ways to communicate the information, such as developing maps and other visual tools, and that there were difficulties around balancing the urgent need to deliver the information quickly with confidence in the information being provided. The landslide bulletin was adapted over a much longer timeframe outside of operational use, allowing the producer team to experiment with different types of content, and different ways of presenting the information.

Team members producing the cyclone bulletin received feedback via email, a method that was felt to be useful and appropriate

during an active emergency response. The feedback was straightforward to incorporate, enabling the bulletins to evolve. The frequency of requested adaptations decreased over time, from multiple emails per day to perhaps only one email per day, as the bulletin better aligned to user needs. When producing bulletins for Cyclone Kenneth (the second cyclone in this case study), the producer team were able to build upon lessons learnt during the preceding Cyclone Idai.

More detailed feedback was provided in a post event debrief, along with two post-event workshops in Mozambique during

which producers meet users for the first time. These debriefs allowed lessons to be learned and captured outside of the time-pressured environment of the disaster response.

For the landslide project, the bulletin went through multiple iterations (Figure 2) based on feedback and discussions between producers and intermediaries (who had previous experience of early warning communication), and key users. Interviewees found that the process of seeking out and incorporating in-person user feedback was useful in strengthening relationships

between users and producers, informing users of the science behind the forecasts, as well as for improving the usefulness and

comprehension of the bulletin for users. Feedback improved producer understanding of the required content and flow of information between institutions at sub-national level.

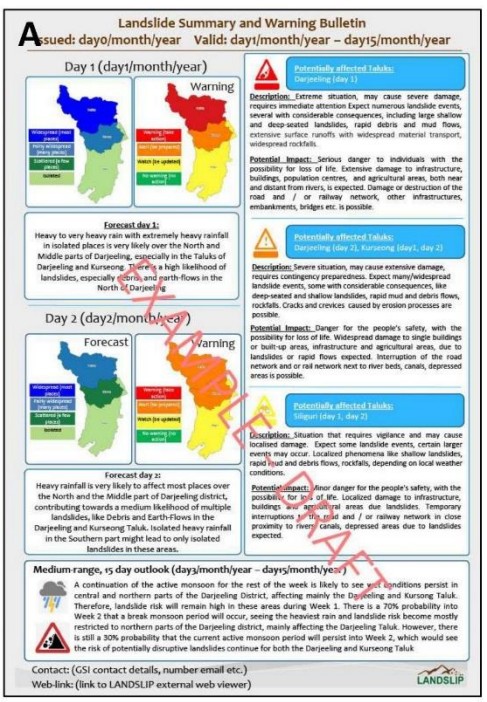

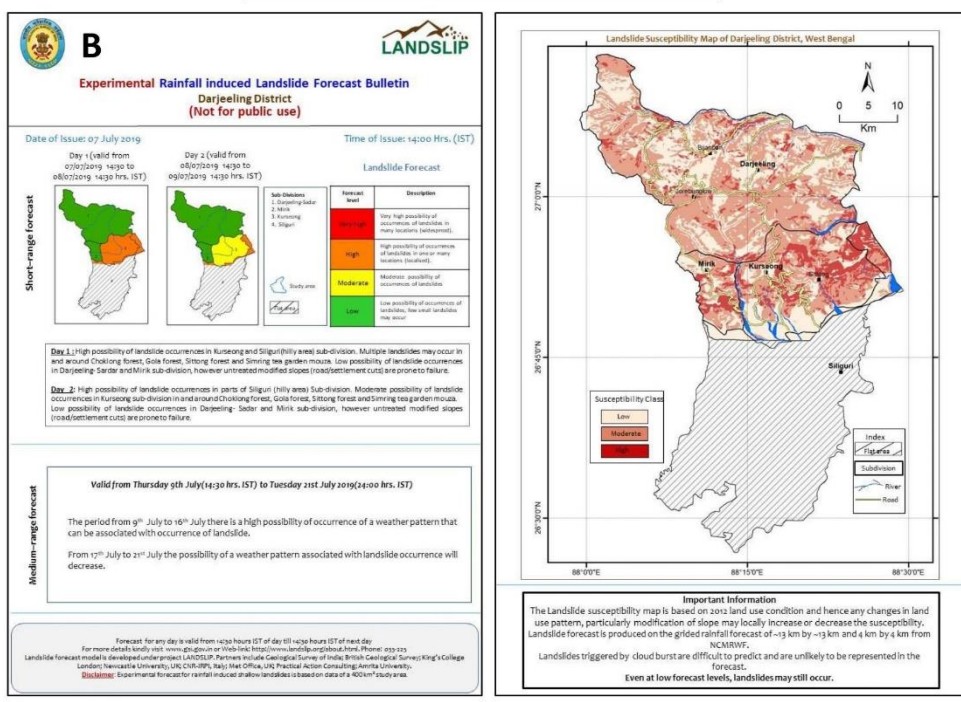

**Figure 2 Examples of prototype landslide forecast bulletins for Darjeeling, India, produced as part of LANDSLIP's experimental regional landslide early warning system (A) January 2019 prototype version, (B) June 2019 prototype version (LANDSLIP, 2019).**

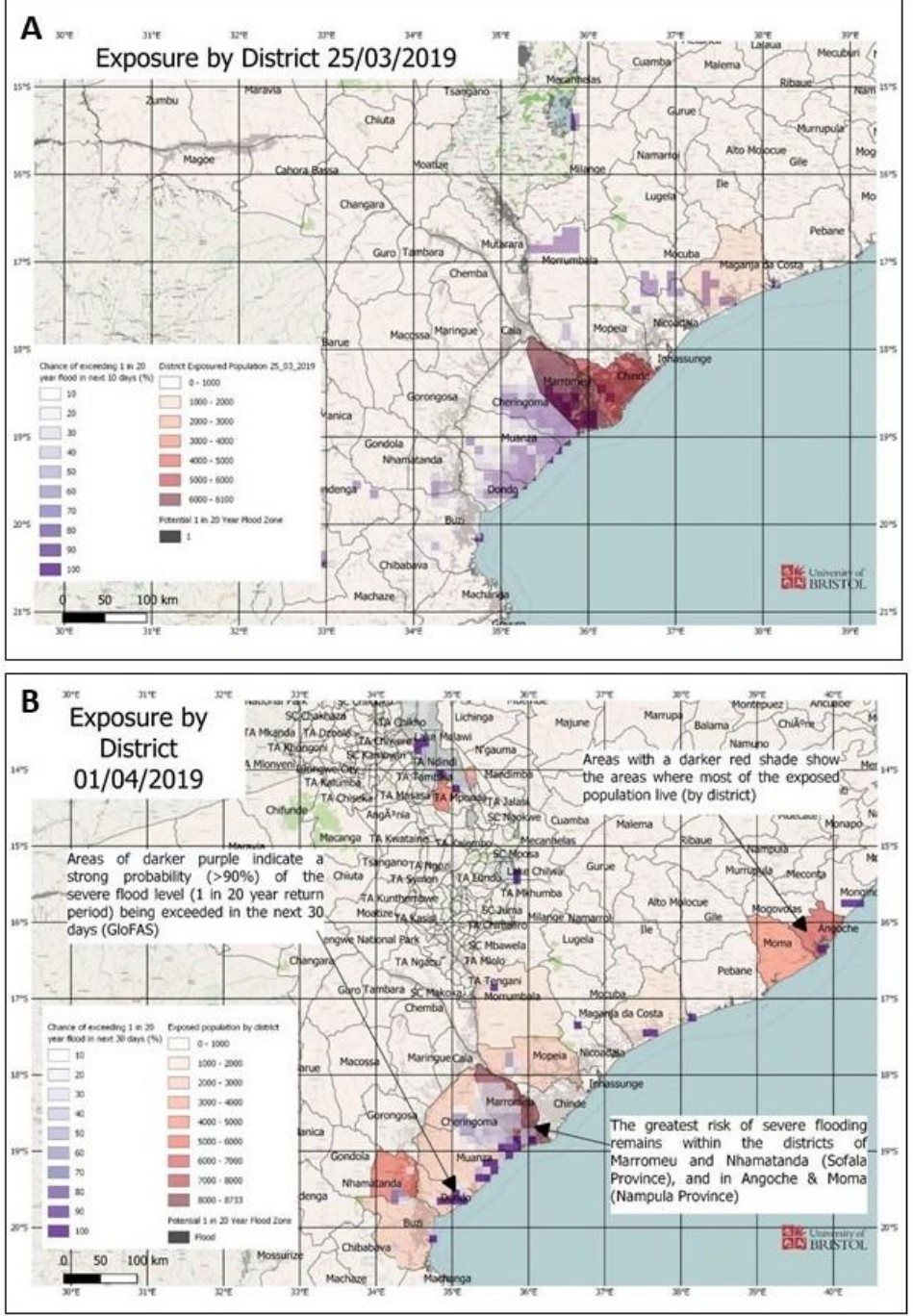

**Figure 3 Changes to cyclone bulletin figure showing exposure by district for Cyclone Idai between the report issued on 25th March (A) and the one issued on 1st April (B); note the addition of text to support interpretation (University of Bristol, 2019).**

## 5.2 Bulletin content

As explained previously, the bulletin content evolved iteratively over time. Decisions were made to include, exclude, or alter the content from the first version developed. This section will cover some of the key changes in the content and discussion points around what to include in a bulletin found in the interviews, changes observed in the bulletin versions, and discussions.

> "I think the biggest pressures were in making sure that the key information that we included was really the most important information for the users who were going to be making the decisions on the ground, and making sure that we were interpreting the forecasts correctly in such short periods of time, that was very stressful." – Cyclone project interviewee 1.

Table 3 summarises some of the key features and changes to the cyclone and landslide bulletins, which are described in more detail in the following sections.

**Table 3 Key features and changes to the content of the bulletins, including layout, text, visuals, and information.**

| Content | Both | Landslide bulletin | Cyclone bulletin |
|---|---|---|---|
| Layout | Summary information at the beginning. More detailed information provided later. | Evolved from 1 page to 2 pages. First page providing changing information, second page containing static information. | Cyclone Idai bulletins evolved from 9 pages to 13-15 pages, as new information added. Cyclone Kenneth bulletins evolved from 5 pages to 10 pages. Summary information as bullet points on first page. Update section added, summarising changes since last bulletin on first page. |
| Text | Simplification of terminology. Reduction in the amount of text provided. Text accompanying visuals to explain them. | Text descriptions of each day's forecast provided instead of levels of warnings. Changed title from "warning" to "forecast" and "experimental" added. Forecast level terminology changed from [Widespread (most places); Fairly widespread (many places); Scattered (a few places); and Isolated] to [Less likely; likely; more likely; most likely] then | Summary first page layout edited to be easier to read. Methodology section removed (remained available as static information) |

| | | | |
|---|---|---|---|
| | | to [Very high; High; Moderate; Low].<br><br>Terminology explanations provided in key. | |
| Visuals | Labelling of key places (particularly if mentioned in text) and administrative areas onto maps.<br>Increase in the number of visuals (maps and graphs) with keys and supplementary text. | Removal of weather forecast maps and focus on landslide forecast maps.<br>Forecast key colours changed to IMD traffic light colour system.<br>'Spots' of colour added to maps where warning level is higher/lower than assigned administrative level. Changed to freestyle shapes.<br>Landslide susceptibility map and text included on second page. Changed to greyscale, then to red tones. | GLoFAS colour scheme changed to traffic light system. Map of area focusing on added to first page.<br>Various maps and graphs added: flood hazard map; graph of temporal forecasts from ECMWF; probability of exceedance of severe flood level; timeline of observed flood extent maps<br>Satellite imagery maps added then removed.<br>Simplification of graphs and maps. |
| Information | Evolving content of type of information.<br>No advice included. | Warning, vulnerability, impact and action content removed.<br>Important information section added to second page with information on uncertainty and caveats/limitations.<br>Added disclaimer in red text below title.<br>Rivers and roads added to static maps. | Evolving to include three main pieces of information: 1) meteorological forecast; 2) flood forecast; and 3) flood hazard and population exposure information. |

### 5.2.1 Layout

The forecast bulletins produced for landslides and cyclones followed a similar structure where summary information is first provided with key points, including updates and changes to the situation, with more detailed information in the following pages for responders who required that in-depth content. This summary information emerged in both the cyclone and landslide

bulletins as a key piece of learning about effective layout.

"We needed this front page that had kind of the key highlights and the main points that we needed to get across in the bulletin that could be read through really, really quickly by people who really didn't have time to be reading all of this information and looking at hydrographs and that sort of thing." – Cyclone project interviewee 1.

The landslide bulletin followed a two-page structure, where the first page provides information on current forecasts (this page changes daily), whilst the second page provides static information on landslide susceptibility and important information on the uncertainties and limitations of the information provided (Figure 2). That way, the first page provides up-to-date big-picture forecast information and the second page provides higher resolution information that can help to enrich the lower-resolution forecasts. Similarly, for the cyclone bulletin, it was useful to have summary information at the beginning of the bulletin with more detailed information in the following 10-15 pages.

**5.2.2 Text**

User feedback led to a shift away from the use of technical terminology in both projects. Interviews highlighted the importance of feedback from users, intermediaries, and a multi-disciplinary producer team, bringing in a range of different perspectives and backgrounds to help to identify language which was too technical, insufficiently explained, irrelevant, or confusing for users. For the cyclone bulletins, the intermediary was able to look at the terminology from the perspective of humanitarian users, and for the landslide bulletins, the inter-disciplinary nature of the project (including intermediaries within the team) meant that colleagues across the consortium could contribute to the refinement of the terminology.

"So, initially, we were using quite a technical description, so we had probabilities of exceedance, and all these kinds of things in there, and in the end I think we had, basically, the wording was kind of, "severe flood", or "worst case flood" and these kinds of things, and actually, since the reports we have thought of further ways to simplify the explanation we're presenting, and if you look at all the reports we would now actually produce something that's quite different to all of them. So that really did evolve." – Cyclone project interviewee 2.

Both inter-disciplinary teams experienced challenges relating to different understandings and uses of key terminology. Relatively subjective terms such as "exposed", "vulnerability", "risk", "susceptibility", and "affected", for example, were used in various ways by the range of physical and social scientists collaborating in both case studies. The importance of developing a shared understanding and standard lexicon for the bulletins, which would be understood in the same way by everyone producing and using the bulletins, was highlighted. The emergency context within which the cyclone bulletins were developed created specific challenges in developing this shared understanding with limited time and remote communication, underscoring the value of preparing templates and agreed terms and definitions in advance (see section 5.8 on preparedness and protocols). The landslide bulletin project context developed this shared language through an evolving process: initially there was often confusion between project partners and frequent discussion and disagreement on terminology; this evolved into awareness and understanding that points of conflict often related to the different understanding of specific terms across different disciplines. A formal activity captured these differences which then developed into a shared lexicon for the project to use going forward.

"Even though we are all working on this LANDSLIP team now, it seems that also the different disciplines had a different understanding about things. There was a big discussion about the word "susceptibility", how people understand it. This was interesting for me because it shows that different disciplines use words in different ways…So, it got better over the project, definitely, but in the beginning I think that was one of the big challenges, just to come together and find out that we are not talking about the same thing." - Landslide project interviewee 11.

In both case studies, there was recognition of the need to be careful in the use of words that could affect how the information is interpreted and used by decision-makers. A key discussion point in both cases was around whether advice or warnings should be provided in the bulletins. Both cases avoided terminology which would take the bulletin beyond its use as a scientific forecast information product, and into providing warnings and advice to users (see section 5.2.4 on exposure, vulnerability and impact information), however it should be noted the inclusion of advice terminology is an ongoing and open discussion point within the landslide case study.

"When we're using [Global Flood Awareness System (GLoFAS)], we have to be very aware of the fact that it is not producing official warnings, so we don't tend to ever use the words 'flood warning' from GLoFAS, we'd always talk about it in terms of forecasts, because it's not a national forecasting centre, it's not responsible for providing warnings and we don't want to imply that." – Cyclone project interviewee 1.

The landslide project initially changed its terminology to align with the India Meteorological Department's (IMD) terms, after agreeing that the comparative terminology (e.g. "more likely") in the bulletin's first iteration was likely to be confusing for users. However, concerns about IMD's terms being warnings and advice about actions to take (e.g. using the term "watch") led to further revision and the team identified alert levels (e.g. "low" level to "very high" level likelihood of a landslide occurring) as the clearest terminology.

### 5.2.3 Visuals

Both teams made numerous changes to the visualisation of information during the iterations that the bulletins went through (Figure 2 and Figure 3). Changes were made as understanding about what information was most useful and how to communicate it effectively developed, incorporating feedback from intermediaries, users, and producers. Key changes related to the content of maps, the use of colour, and the use of supporting text to improve the effectiveness of visualisations (Figure 2 and Figure 3). A key change in the cyclone bulletins was also the simplification of visualisations, such as graphs, that were too complicated for users to clearly understand. The landslide bulletin team also removed weather forecast maps and focused instead on only having landslide forecast maps in the bulletin to avoid confusion between two 'forecast' map versions.

Both bulletins use maps to convey forecast or impact information. Rainfall forecast maps were added within a few iterations of the cyclone bulletin; however, the quantity of maps was streamlined as the bulletin developed to simplify the information being provided and provide the most relevant information to users. Additionally, the maps were developed to be consistent

with the text in the bulletin, ensuring that every location which was mentioned in the text was manually marked on the accompanying maps. The team began to superimpose different maps in later versions of the bulletin to include the layers of information needed.

For the cyclone bulletins, satellite observation information maps were requested by the intermediary, but the producer team decided not to include them. The team did initially try to incorporate this information, but found that the satellite observation data did not add any value to the other information due to the resolution the satellite observation maps could produce. This decision raises an important consideration regarding perceived user needs and balancing those with what is scientifically possible given data constraints.

"[The intermediary] was quite keen to have satellite information…[it] was very time consuming and actually didn't match the scale, so I did it a couple of times and then I just thought it was a waste of time because it doesn't bring any extra information, so then we agreed with [the intermediary] that we would not even look at it for [Cyclone] Kenneth, and they were OK with it." – Cyclone project interviewee 3.

The use of colour evolved over different versions of both bulletins. The cyclone bulletins initially adopted the colour scheme used by the Global Flood Awareness System (GLoFAS), but changed to a traffic light system to better and more clearly illustrate the varying levels of risk within the maps. The landslide bulletin adopted the India Meteorological Department (IMD) traffic light colour scheme for their landslide forecast levels to align with what users had familiarity with (Figure 2). The colour scheme for the static landslide susceptibility maps were subsequently changed to a different colour scheme (greyscale, then changed to shades of red) to avoid confusion between forecast and susceptibility information (Figure 2). Only one interviewee mentioned specific potential adaptations to enhance accessibility, with consideration of the needs of users with deuteranopia (colour visual impairment).

Both teams ensured that visualisations were accompanied by simple text explaining what was being presented (Figure 2 and 3). Given the complexity of the information being provided in the graphics, and the range of possible interpretations of visual information, explanatory text was deemed essential by producers and intermediaries (and from user feedback) to enable users to understand the context and meaning of the maps and colours in the bulletin.

### 5.2.4 Exposure, vulnerability and impact information

The way key assets are labelled and identified evolved through the various iterations of the cyclone bulletin, with key locations and features such as towns, roads, and other major infrastructure being clearly labelled. As the bulletin went through different iterations, the producer team prioritised addition of important locations (dams, roads, rivers and towns), providing valuable context for the forecasts. However, in areas with low availability and quality of open access, online map coverage this information was extremely challenging to collect. Interviews with cyclone bulletin producers highlighted that the more detailed the bulletins were, the more effort it took to update them on a daily basis. Labelling locations and assets was a manual process

in these bulletins, as there were no locally-specific maps or databases available to the producers to automate this process. Some assets were labelled specifically at the request and direction of users via the intermediary.

> "[The intermediary] told us there with a dam in the region that people were really, really worried about, which isn't modelled in the GLoFAS flood forecasting system, so we were able to comment on that in the bulletin and then we received feedback that was something that was really useful." – Cyclone project interviewee 1.

> "It was very difficult actually, something as simple as mapping actually became quite an issue, became a real problem, even things like local names of rivers and stuff like that, and how we linked that back to our forecasting system was very, very challenging. It's not actually as simple as just going on Google maps and just having a look at the river names, it's not as simple as that…we do not have local information everywhere in the world, of course, so putting yourself into any region that you literally have no knowledge about is very difficult." – Cyclone project interviewee
> 5.

The landslide bulletin interviewees highlighted that it is extremely difficult, and requires significant amounts of data, to assess and model the exposure of assets and infrastructure to landslides, and the potential impact a landslide might have on them. A landslide may affect areas that are identified as being at high, medium and low risk depending on the type of landslide and the way in which the surrounding environment interacts with the movement of the landslide. Interviewees also stated that the pace
of development in India means that information about assets such as roads which are exposed may quickly become out of date. The team therefore decided to include more detailed information within the supplementary text to accompany the maps and encourage users to supplement with their own information to understand better which assets and infrastructure might be affected locally.

One of the first and most significant changes to the landslide bulletin was the removal of vulnerability information which was
505 included in one of the earliest iterations of the bulletin. There were discussions around using a relative ranking of vulnerability, but these were quickly dismissed by the project team due to concerns and issues regarding the robustness of this methodology, capacity and responsibility for keeping the data up to date, and ethical reservations about providing vulnerability information that could influence the allocation of resources in a way that could drive additional risk and vulnerability.

> "If you provide vulnerability information alongside forecast information, it will skew perceptions and decisions
> around where to allocate resources and where to respond to first, which can be inherently flawed, particularly because the vulnerability data is not updated. It's suggesting something from a scientific perspective that [the forecast producers] don't have the evidence to back up. If they had all the data, it would still not be [the producers'] role to provide vulnerability information to the [users]. It would be the [District Authorities'] role to compare the landslide hazard forecast with their own vulnerability data and their understanding, but it's something massively outside [the
> producers'] responsibility and expertise." - Landslide project interviewee 1.

Both projects addressed changes to the bulletin which were requested by the user related to the inclusion of impact information. One of the main topics of conflict facing the cyclones team was in relation to the estimated numbers of affected people. From the intermediary's perspective, it was important to emphasise the gravity of the situation overall in order to generate sufficient responsiveness and action coherent with the needs on the ground, whereas the team generating the information were focused on providing scientifically-based confident assessments of the number of people who would be affected by a specific facet of the crisis, namely fluvial flooding. As a result, there were very different figures being reported by the bulletin (which focused on the impact of fluvial flooding, from rivers, alone) compared to other sources such as news outlets, and humanitarian and government agencies (which included people affected by the overall hazard). This led to concerns from producers and intermediaries that the discrepancies in numbers would confuse users because they were referring different elements of the risk.

For the landslide project, changes related to the inclusion of impact information have been central to discussions as the bulletin has developed and evolved, with different perspectives across researchers and intended users as to whether and how to incorporate this. Users and some project members requested information about the impacts of the forecasted landslides. There were a range of discussions within the project team as to whether this was possible or appropriate, with the final decision to not include that type of information at this stage, but to encourage the use of the landslide susceptibility map in the bulletin, and other available supporting information for users to support their decision-making. Producers within the team highlighted issues around including impact information that was either too general to be useful for making decisions, or that risked misinforming decisions due to the uncertainty of the information based on poor data. There were alternative perspectives within the group that the inclusion of example impact information would be useful for decision-makers as illustrations of potential damage caused by landslides to support preparations of users.

> "Impact based warnings [are] quite problematic. The problem is, at the moment, impact is two steps forward, we are not there, I would say. First of all, for the impact side, they need more information, other information, that they do not have." – Landslide project interviewee 11.

### 5.3 Information versus advice

Both projects faced decisions about the role of the bulletin in providing advice to users. For the team producing the cyclone bulletins, a key barrier to the inclusion of advice was the fact that the bulletins were being produced remotely, and they did not have direct knowledge or experience of the response or the users, and therefore felt providing advice was beyond their capacity and scope. The landslide bulletin team members reflected on the need to balance the expectations of users to provide advice with the uncertainty and low-resolution scale of the available forecast, the difficulty of providing useful and tailored advice to a potentially wide range of users in a short product, the capacity of users to take action, and who is officially mandated to provide advice. Each landslide bulletin includes a section highlighting the uncertainty and caveats of the information provided, an element that was important for the producers to communicate the confidence and limitations of the forecast information provided.

"We didn't provide any advice on what should be done based on the forecast, because we weren't on the ground. We didn't feel we were in a place to provide that kind of advice….in the actual bulletins it was very much just focusing on the forecast information, what was happening with the cyclone, where was likely to be affected, what the hazards were likely to be, but we left it at that in the bulletins." – Cyclone project interviewee 1.

"When we start moving, again, down the chain and saying, 'This is the kind of action you should therefore take, because we think that this hazard will lead to this type of impact, so you should do this,' - that obviously requires a huge amount of stakeholder engagement, because to be able to even start suggesting what kind of actions should be taken, you need to have an understanding of the capability of people to do something…these are gradually increasing levels of complexity, and realistically, we're only at step one, really." – Landslide project interviewee 10.

## 5.4 Communicating complexity

Both projects focus on hazards that are extremely complex. Following Cyclones Idai and Kenneth, affected areas were at risk of a range of associated hazards including river floods and storm surge. The cyclone bulletin focused on risk of flooding from rivers, so there was a need to contextualise that focus within the bulletin and highlight to the users that the bulletin was not a comprehensive assessment of all hazards associated with the cyclones. The landslide project identified several challenges related to the complexity of landslide hazards and of communicating complex risk in a way that is simple, clear, and understandable to different users. A key tension in the landslide project was between a desire to simplify complex risk information, and the need to avoid oversimplification which could lead to decisions being made based on flawed understanding of the risk and uncertainties.

"We knew that our bulletins were focusing only on river floods, and we specified that in the bulletin. We mentioned, even from the beginning for [Cylone] Idai, we mentioned that there was also storm surge risk, we mentioned even height of the storm surge, and other reports [that were] producing a more multi hazard record." – Cyclone project interviewee 4.

"If [the producers are] going to simplify they want to be really, really clear on what that actually means and have that backed up somewhere that makes sense. Because the risk of them taking on the responsibility for simplifying the information and then the [user] interpreting that, if that goes wrong the [user] then will blame [the producer] for not providing the right information in in the right way. So there is a balancing act." – Landslide project interviewee 1.

## 5.5 Team roles and skills

Both project teams included a diverse range of relevant expertise which many interviewees highlighted as enabling the work to be effective and impactful. Team members brought expertise from different fields, as well as experience of operational forecasting, and skills in science communication which were central to the success of the work. Interviews highlighted the importance of bringing together physical and social scientific expertise, and ensuring that these roles informed and supported

each other rather than working in isolation. In both projects the producers or forecasters had experience of working in an applied context, or had experience of science communication, which was seen as important by the interviewees.

> "When I talk about issues I mean, on one hand, you need the experience of someone to understand the models and how to interpret those and write something that's factually, scientifically correct, but then you need experience from somebody who understands the end user to know how to translate and interpret that into something meaningful." –
Landslide project interviewee 7.

The interviewees from both case studies identified the roles and skills which were instrumental in delivering the bulletins effectively. Critical skills included: understanding forecasting models, their limitations and outputs; understanding the technical operational requirements for generating the information for the bulletin; contextual knowledge; and understanding of how to effectively communicate the information. Key areas that were highlighted as being gaps in the production of cyclone
bulletins that would be beneficial to future work included: a role for a representative of the target community or user group; the need for operational forecasting skills; the need for a clear and structured approach to assigning roles and tasks in line with availability and capacity (see also section 5.8 preparedness and protocols); and the need for redundancy to ensure that roles can be fulfilled in the absence of any individual.

Landslide bulletin interviews emphasised the importance of the role of understanding the science that underpins the model,
including an understanding of weather forecasting, meteorology, geology and geomorphology, and geological and geomorphological engineering. An understanding of the model itself, including its limitations, caveats, assumptions and uncertainties, was also emphasised. It was noted that running the model will additionally require skills in coding, information management, maintenance and repair, as well as computer science skills in software development, alongside analytical skills to interpret the outputs of the model.

Interviewees also reflected that a key piece of learning in the landslide project had been around the importance of expertise in social science, as well as an in-depth understanding of the local context, in order to design a bulletin which is understandable and useful. Interviewees reflected on the wider project teams' evolution in recognising the importance of effective institutional mapping for the success of the project – a process that provides an understanding of the complex networks of stakeholders, relationships and decision-making opportunities the bulletin feeds in to. The project team concluded that institutional mapping
was a key priority at the outset of the project, as well as the importance of a continued prioritisation on ongoing institutional mapping as the purpose and users of the bulletin evolves or the institutional landscape shifts.

### 5.6 Priorities and relationships

In both contexts, the production of the bulletin involved a range of stakeholders with diverse responsibilities, mandates and priorities, presenting similar challenges for the teams developing cyclone and landslide bulletins. For the landslide project, the
610 central conflicts related to different expectations about what the project set out to achieve and what was possible to deliver.

The expected producers of the forecast information (which was still under development) faced pressure to deliver an operational early warning system, given the ongoing landslide risk context the project was working in. In contrast, the physical scientists within the project team cautioned on the need to ensure the bulletins are based on robust and sound physical science that has been tested, evaluated, and validated thoroughly. There were ongoing discussions and nervousness particularly from the physical scientists within the team regarding releasing bulletins beyond the project team, even when steps were taken to ensure it was clear the bulletin forecasts were untested and should not be used to make decisions or take actions based on them.

> "Our initial scientific scope was very much research-oriented, with the idea of having the expert users involved in the development. It's more difficult when your scientific scope doesn't necessarily match up with what your non expert user is anticipating having at the end." – Landslide project interviewee 10.

In both contexts, the bulletins were produced by teams working across different locations from a range of institutions with different responsibilities, priorities and institutional cultures. Across both contexts, the interviews found: a recognition of the challenges of working in a multi-disciplinary team formed across institutions and locations; a significant amount of positivity about the collaboration and about how challenges had been overcome; and that working patterns and relationships had improved over time. It is worth noting that in these case studies, the bulletins were produced in English, with stakeholders who were all accustomed to using English as a working language. In other contexts, linguistic barriers will need to be considered as a potential barrier to effective collaboration.

The cyclone bulletin producers reported that communication across locations was a key difficulty, as different components of the bulletin were produced by three discreet producer teams in different locations. Two of the producer teams were able to share an office space whilst producing the daily bulletin, whilst the third producer team and the intermediaries were unable to physically join them in the same space. They reported that those who shared offices were able to communicate in person, sharing information and updates, and collaborating and coordinating to generate, interpret and convey information, while communication with other team members based at a different institution was reliant on email.

> "I think what went particularly well was the collaboration throughout, I was really impressed, actually with first of all having [some of the producing team] being in the same place physically, it really helped. It meant that we could have discussions and talk about things and review things very quickly." – Cyclone project interviewee 5.

> "Pretty much everything was over emails. We didn't even have time to set up basic things like how we would share the information, so that was a bit of the challenge…But at the beginning it was very much two separate groups and two separate records. At the end you could actually see that the group was very much working as one." – Cyclone project interviewee 6.

For the landslide project, physical distance was also a major issue – with team members located across India, Italy and the UK and from a range of institutions. However, again, the longer timeframe and funding for travel embedded within the project plans meant face-to-face meetings were possible at strategic intervals within the project timeframe. Major developments in bulletin content and changes typically occurred during scheduled in-person visits between team members, with slower, more incremental changes observed in between these visits, using email and video conferencing calls to discuss and share.

A key factor which benefitted the production of the bulletins for Cyclones Idai and Kenneth was the collaboration between the different institutions as well as with the intermediary. Team members highlighted the value of existing relationships, particularly between intermediaries and two of the producer teams. The landslide bulletin team reflected on the importance of developing relationships between consortium members over the course of the project, and the value of the bulletin in providing a focus for integration across the wider research project. Limited pre-existing relationships within the landslide project meant that consortium members had to spend time in the project learning about each other, the different experiences, capacities and ways of working between the institutions.

## 5.7 Accuracy and evaluation

In both contexts, there was tension observed between what the users of the bulletin wanted, and what the producers of the bulletin could provide. This gap manifested in two key areas: firstly, what is possible for the science to produce in terms of spatial and temporal detail and certainty; and secondly, what the producers of the bulletin felt they could state with integrity.

> "One thing that [the intermediaries] keep saying they wanted and that we haven't provided to them, and it would be terribly difficult to provide to them, is an estimation, an evaluation, a validation of the thing…because such a thorough scientific validation would require observational data on the ground that we don't have." – Cyclone project interviewee 3.

> "[The producers] keeps asking us, "What are the uncertainties? How do we communicate [discrepancies in the forecast information]...?"… at the moment we don't have the scientific back up, because we haven't been able to do an evaluation of the product … we don't fully understand the uncertainties in the model that's been done." – Landslide project interviewee 10.

Both projects experienced challenges in the gap between what scale, detail and accuracy the users wanted to be able to make better decisions, and what the available data, science, and forecast technology (and project scope) was able to provide. For example, while providing forecast information in response to Cyclones Idai and Kenneth, the team identified issues with using global systems to generate local information; the intermediary wanted more information at finer scales and longer lead times than the science could provide. Both projects reflected on challenges in needing to manage user expectations regarding the level of detail and certainty that is possible to provide.

Both teams reported significant challenges in dealing with uncertainty and validation or evaluation. A key challenge in the landslide project related to the scale at which it is possible to provide information (with the available data and resources) and the varying levels of probabilities within that spatial unit, and the level of certainty that decision-makers need regarding the likelihood and severity of possible landslide events. In the case of Cyclone Idai and Kenneth, the teams found the expression
of uncertainty to be a key challenge as they worked to deliver the bulletins. There was a continuing effort to balance the users' need for high levels of certainty in the information provided with what could be said scientifically. In expressing the level of uncertainty, the wording changed as the teams worked to understand how much information the users needed, for example about the probabilities of different thresholds being exceeded, and how to accommodate the uncertainties around factors such as the amount of water in rivers, the topography of the surrounding areas, and the population.

Both teams also reported challenges related to the gap between the need to validate or evaluate the accuracy of the model predictions against what was being observed, particularly to be able to better estimate the uncertainty, skill and accuracy of the forecasts, and the lack of time or data to actually carry out this process. Both the cyclone and landslide teams discussed in interviews the issue of validation, highlighting the importance of verifying the performance of the model against the actual events in order to determine how accurately events were forecasted, and working with users to evaluate how useful the bulletin
was in supporting effective humanitarian decision-making and response.

For the cyclone bulletin, there was no time to collect data or evaluate the model predictions. The lack of evaluation of the flood forecasting model also presented challenges for the bulletin itself as it necessitated the communication of uncertainty into a context where the users required concrete information (see section 5.7 understanding users). The landslide project identified similar challenges to the cyclone bulletin even in a non-emergency context. Landslide bulletin producers reported during
interviews that until the model's skill is evaluated, it is not possible to understand and properly use the model outputs to provide information about risk for the users that is reliable, which reinforces the tension between what users expect the bulletin to deliver, and what is feasible within the timescale.

## 5.8 Understanding users

A major challenge experienced by the team in rapidly developing bulletins to guide the response to Cyclones Idai and Kenneth
was the lack of understanding about who the users of the bulletin were and what their needs were. Due to the constricted time available to produce the bulletins on-demand during an ongoing emergency, and the pressures of the users to respond to the impacts of the cyclones, the interaction between producers and users was carried out by the intermediary. Team members highlighted several issues this raised: they were unsure what the users would be using the information for, and how far in advance they needed the bulletin. They were unsure as to how understandable the language was, both in terms of English
language fluency and technical language. They did not know how much detail was appropriate to include, and how far users were familiar with technical concepts as well as terminology, and how to effectively communicate the forecasts including their uncertainty. As well as the backgrounds of the users, the team was unaware of the resources users had access to in terms of software packages, which affected how they presented visual information such as maps.

"We felt we probably had a reasonable idea of what we would do and how we would present it, and then I think the
process of actually doing a bulletin has told us that we actually had no idea who the end users actually were or what
they really wanted." –  Cyclone project interviewee 2.

Over the course of the process of developing and refining the cyclone bulletins, this understanding improved and was identified as a valuable piece of learning. Interviews found that the team's perspective of what information was useful to provide changed over time as they became more familiar with user needs (through feedback from and via intermediaries), and also highlighted some learning about how to improve and expedite that development in future work. One key example that reflected how this understanding developed was the inclusion of flood recession information, which the teams had not expected to be useful for the users, but which was valuable in making decisions about where and when to direct response support.

Landslide bulletin team members similarly reported challenges at the beginning of the project relating to clarity of who the user of the bulletin was intended to be, specifically whether the target was a technical expert user, or non-technical policy and practice users, or the general public, and what this meant for the type of tool or product that would be useful for different users. However, the team was able to improve this knowledge over a greater period of time, and use this knowledge to adapt the bulletin content for the users as they were identified and better understood.

**5.9 Preparedness and protocols**

Both teams recognised the importance of preparedness, protocols, training, and templates to ensure the information provided in an emergency context is streamlined, reduces decision time, reduces error, and ensures consistency and sustainability of producing forecast information in the future. Both teams agreed on the value of having protocols and processes in place to guide and support effective communication, coordination and collaboration across the different institutions, ways of working and models. The importance of training was also discussed during interviews to ensure that the bulletin producers would always have the knowledge needed to interpret model outputs and find key information, and to have those skills in place and established prior to an emergency situation. The two case studies adopted different approaches to sustainability. The bulletins for Cyclones Idai and Kenneth were developed to provide additional bespoke forecasts for those specific events, without a focus on embedding them within national institutions or sustaining them into the future. The bulletins for the landslide project were always intended to be sustained long-term, with the bulletin co-developed with the key national institutions with responsibilities for longer term application.

The emergency nature of the bulletins developed for Cyclones Idai and Kenneth presented unique challenges in terms of coordinating across teams in the absence of protocols, rotas, and the availability of team members. Because there was no dedicated team in place with the specific responsibility of providing this service, it was necessary to pull together individuals at extremely short notice and to assign responsibilities according to who was available and had the capacity to be involved, with no clear indication of how long the work would continue for or formal coordination of inputs and tasks. Additionally,

interviews highlighted the value of having a prepared template in place to guide the structure and content of the bulletin for future work and save time spent in exploring the most suitable ways to present the information in terms of layout.

> "The product that we were able to deliver together…was the best that we could do within the time, but everybody on both sides would agree that if we had sat down, if we had the opportunity before the event to sit down and design it, we would do things differently and we'd be able to have a much superior and robust product, that was of course tested
a little bit more, and we would have had some confidence." – Cyclone project interviewee 5.

The landslide project was able to utilise the longer timescale of the project to develop Standard Operating Procedures (SOPs) to guide the process of issuing warning information in the form of a bulletin. The SOPs for the production of the bulletin cover when and how to generate the bulletin, and when and who to share it with, as well as how to use phrasing, colour coding, and maps to communicate the information effectively. A standard template and library of phrases are available for any production
team member to access – thus ensuring redundancies are built into the system and ensuring legacy and sustainability long-term. A key challenge for the longer-term, post project, producers of the bulletin is the issue of re-assignment of staff, so the SOPs need to be usable for new team members who have not been involved in the co-production process in order to have a sustainable legacy and impact. Additionally, sustainability will require familiarity and confidence of the users in the information provided in the bulletin and their own responses to it.

## 6 Considerations for developing forecast bulletins

The bulletins evolved to include content that is consistent with best practice identified in the literature on risk communication. There were similarities between the cyclone and landslide case studies in the consideration and use of visualisation, maps, colour, text to accompany images (Visschers et al., 2009), identifying key locations, careful selection of words, avoiding jargon, clarifying common terminology (Anderson-Berry et al., 2018; Robbins et al., 2019), prioritising information useful to
make decisions, simplifying complex information, providing summary information up front and more detail later, and communicating uncertainties (Patt and Schrag, 2003). The main differences between the landslide and cyclone bulletin content was the length.

In their previous evaluation of the cyclone bulletins, Emerton et al. (2020) discussed the challenges and lessons learnt from the process, drawing out recommendation relating to both the bulletin production and dissemination, and the development of
the underlying scientific models and data. There is inevitably (and reassuringly) some overlap between their recommendations and the ones made here. However, here we use our unique analysis from the perspective of the people involved in producing the bulletins to focus on the key considerations to follow in developing a bulletin from scratch to provide forecast information for decision-makers. It is hoped that by drawing out some of these considerations and discussions, this analysis will provide an opportunity for others involved in forecast production for natural hazards to learn from these experiences and to work
towards collaborative solutions to some of the challenges.

## 6.1 Engaging with users

Despite much of the literature emphasising the need to engage with users and the benefits of co-production of information and resources (Kox et al., 2018; Robbins et al., 2019; Gill et al., 2008; Carter et al., 2019; Lemos et al., 2012), both case studies had limited direct engagement with users in the evolution of the bulletin, particularly from the beginning stages. The reasons were different for each project. The cyclone case study had limited engagement with users during the active issuance of bulletins due to time pressures and humanitarian response needs restricting access to users – the intermediary role took the lead on communication between groups. However, the engagement between producers and users improved after the cyclone events were over, when workshops were held between producers and users to discuss the experience and find ways forward.

For the landslide case study, users were not directly engaged in co-production of the bulletin from the very beginning of the project, with their involvement increasing in the middle and latter stages of the development of the bulletin, once the research team had clearer understanding of what forecasts would be possible, and therefore who the users would be. There were different perspectives within the team, and between physical scientists and social scientists, on the timing of when to involve users. Social scientists within the team highlighted that involvement of users from the beginning could enhance the users' understanding of the limitations of the models and data and their inherent uncertainties (Frick and Hegg., 2011; Anderson-Berry et al., 2018; Taylor et al., 2018; Bica et al., 2019). Institutions responsible for providing forecasts to government authorities felt pressure to begin sharing forecast information as soon as possible (Patt and Gwata, 2002). The engagement with users scaled up when the science had progressed to be well-enough understood, and also in response to institutional pressures within the producer team. However, many of the physical scientists within the team see this sharing of untested knowledge as risky, particularly before validation of the forecast skill has been thoroughly conducted, as there is a risk of users making decisions based on untested science (despite being instructed not to), which could have severe and long-term consequences, as evidenced in real-life case studies and published literature (Patt and Gwata, 2002).

This demonstrates the difficulties and challenges producers face when developing brand new forecast products and trialling new science and trying to apply it at the same time, typically under extreme time pressure. The tension between innovation and application often makes it difficult to directly engage with users from the very beginning. In addition, the contrasting perspectives, approaches, and concerns of physical scientists, social scientists, and practitioners or intermediaries within project teams of this nature is a key challenge. Determining when is "right" to engage with users, and when the "right" time for sharing outputs is, will need to be debated and decided on a case-by-case basis. It is likely that the "right" time will not be universally agreed, or will at the very least stretch comfort levels within an interdisciplinary team. Discussing, understanding differing perspectives and concerns, and collectively agreeing approaches from the beginning can support this process, although it is likely to remain a challenge for all research into application projects (Carter et al., 2019; Lemos et al., 2012).

These two case studies also show that the relationship with users needs to be developed over time, and if that time is not available, then using intermediaries that have existing relationships or knowledge of the context and needs can bridge that gap temporarily, whilst the relationship is built (Carter et al., 2019; Lemos et al., 2012). Identifying the user group was essential

to developing forecast information, and an understanding of their needs and level of knowledge was demonstrated as being vital in knowing what to provide and how to tailor bulletins (Taylor et al., 2018; Morss et al., 2005; Wachinger at al., 2013; Speight et al., 2018; Carter et al., 2019). The institutional mapping and stakeholder engagement activities, often led by social scientists and intermediaries, were essential in developing this understanding of user needs to develop the bulletin (Lambrachet et al., 2019; Robbins et al., 2019; Morss et al., 2019; Carter et al., 2019).

### 6.2 Value interdisciplinary skills

In interview discussions on the skills and roles needed to develop forecast bulletins, there was consensus and emphasis across both case studies that a range of disciplines and skills are needed (Morss et al., 2005; McBean and Rodgers, 2009; McInerny et al., 2014; Anderson-Berry et al., 2018; Robbins et al., 2019). Physical science-related skills and expertise were clearly articulated by interviewees and recognised as being foundational to developing forecast information (e.g. operational meteorology, geological engineering, geomorphology, coding, running, maintenance and interpretation of the model and its outputs). There was also a recognition across teams that social science and intermediary type skills (e.g. communication and understanding of context and users) are important to develop forecast bulletins but interviewees found it harder to articulate discretely the specific skills beyond physical science that they saw as critical.

Whilst there is a lack of published guidance on the process needed to develop a bulletin (Stephens et al., 2015), there is a wealth of existing practical knowledge often gained through experience that was channelled into both projects through intermediaries and applied physical scientists. Both projects relied on an intermediary role within their team to guide the iterative development process. The intermediary roles' ability and effectiveness to guide the process was enhanced by an existing appreciation within the physical scientists' team of the value they add, the importance of good communication, and a desire to provide useful information. Where this understanding and appreciation of added value was lacking to begin with (in the case of some physical scientists), it evolved over time as pressures to operationalise bulletins increased awareness of the importance and complexity of communicating useful information to users.

Further research is needed on what specific expertise, knowledge, experience, skills, training are needed to make up a good 'intermediary' role to inform practical guidance (Cumiskey et al., 2019). Given the consensus on the value of partnership and social science skills to project success, further effort is needed to better articulate these contributions and define specific areas of expertise for future work, otherwise they risk being overlooked relative to the more defined physical science skills. Education and awareness is also needed for physical scientists, users, and funders to recognise the need and value of social science sub-disciplines at a level *equal to* physical science in projects where application is an objective. There will be different perspectives on where intermediary roles should sit, with examples in this research of intermediaries being external to a given project, or embedded within a project team (Carter et al., 2019). Regardless of location, it is important to ensure these roles and skillsets are emphasised, especially in institutions where such applied or social science roles may not be currently prioritised.

Both teams reflected that the operation of a daily forecast bulletin is a full-time occupation, and funding for a team with the required range of skills to dedicate the appropriate time to issuing the bulletin (and developing protocols) needs to be recognised. Education, capacity building, and training is needed – not just for the users – but also for the producers to be fully skilled and competent to operate. Resources, training, and time is needed to achieve this both for producers and users (Robbins et al., 2019).

### 6.3 Meeting user needs

A recurring and evolving theme throughout both studies was the balance between science and user needs, reinforcing established principles for effective co-production (Carter et al., 2019). In particular, there was a delicate balance between providing robust, skilful, and accurate scientific information, and providing information at the resolution, accuracy, and reliability needed to support decision-making (Patt and Gwata, 2002; Lemos et al., 2012). Issues around the desire to validate or evaluate the forecast models emerged, conflicting with limited data availability, time constraints, and the pressures to deliver. Whilst the time available to develop the bulletins varied enormously between studies, the pressure of time was present in both. Notably, the challenge of balancing the science and the needs of the user did not get significantly easier with time.

Echoing the findings of Kox et al. (2018), one of the lessons from these case studies which helped bridge the gap between science and user needs is communicating and working with the user in an open and transparent way to ensure the information and its limitations and uncertainties are clearly explained and understood and to manage expectations (Patt and Gwata, 2002; Lemos et al., 2012).

In both studies, there was limited mention of considering accessibility requirements of professional users. Both bulletins were produced in English language, whilst the national language in Mozambique is Portuguese, and there are several dominant languages in the Darjeeling and Nilgiris Districts of India (Hindi, Tamil, Nepali and Bengali). The main reason for the use of English is likely because the common, dominant language of producers in both projects is English, and English is understood in both Mozambique and India. This aspect was not investigated within the interviews, so it is uncertain whether this was an active or unconscious choice. It is also unknown from the finding whether the users experienced any difficulties with this choice of language. Only one user mentioned consideration of colour vision impairments. This suggests producers and intermediaries assumed users, given their job role in a government or humanitarian organisation, would not experience any issues with accessibility of the bulletin information beyond issues with technical jargon. Similarly, most of the published literature on issues related to digital literacy, accessibility, visual impairments, and language are based on studies of the general public.

Assumptions about the accessibility of the bulletin for users who may have diverse linguistic, educational and professional backgrounds and training, as well as diverse sensory requirements of communication appears to be pervasive and is fundamentally problematic as an unconsciously biased approach to developing bulletins for specialised and/or professional user groups in policy and practice and needs to be further explored. Feedback mechanisms with the users *should* be able to

pick up on any issues the users experience, highlighting the importance of feedback from users, however a more proactive and conscious approach would be better. More research is needed into accessibility barriers and considerations for professional users of forecast information.

## 6.4 Mandates and responsibilities

There were tensions in both studies between balancing science and user needs not only because of what is possible for scientists to provide, but also influenced by tensions related to the mandate and purpose of science and scientists (specifically physical science forecasters), and also by the aims, scope, and restrictions of funded projects. In both studies, there were challenges related to users requesting information that was beyond the scope of the project, for example, the inclusion of exposure, impact, and vulnerability data or assessments which could be used to influence actions that affect people's lives.

The official responsibilities of producing forecast information were different for each project. The cyclone bulletins were produced by non-responsible institutions at the request of a key stakeholder. As such one of their main focus points was to ensure scientific rigour in the information they provided, to protect institutional reputation, but they were not officially responsible or mandated with providing the information - it was supplementary to formal mechanisms and information.

For the landslide bulletins, this was more complex as the project lifetime covered a period when the institution that would undertake production of bulletins beyond the project funding was undergoing a major shift in their institution's role and official mandate during the project lifetime, changing from their previous focus on response to landslides towards the provision of information in advance of landslides. This change in mandate required a significant institutional culture shift and a rapid learning curve to overcome the initial lack of experience, familiarity and confidence in issuing forecast information.

Landslide project interviews highlighted the impact of institutional mandates and responsibilities on the bulletin, emphasising that the producer's responsibility was to provide forecast information, and not to issue warnings. This directly affected the content of the bulletin: the terminology of "forecast" rather than "warning" was carefully chosen, it was decided not to provide (or update) vulnerability information in the bulletin, and it was decided not to provide advice on actions to be taken in response to warnings.

In published literature and real-world examples, there is a tension in not just what science can provide, but whether they should provide it at all. This comes to the fore particularly when science is used to make decisions alongside other evidence (Frick and Hegg., 2011). When these types of decisions are the role and responsibility of government officials, but need to be informed by science, then scientists need to be careful in considering what they provide, how they provide it, and how to communicate it (Kox et al., 2018; Patt and Gwata, 2002). There needs to be a clear and transparent agreement and awareness of the difference in roles, responsibilities, and mandates of the producers of forecasters compared to that of the institutional decision-makers (Sukhwani et al., 2019). This is vital in developing and protecting forecast producer's scientific reputation and the users' trust in their abilities (Carter et al., 2019; Patt and Gwata, 2002).

### 6.5 Developing strong working relationships

The importance of not only having a strong interdisciplinary team with specific skills, but also to have good relationships within that team was strong in both case studies. Developing a shared understanding and lexicon from the beginning (Lambrecht et al., 2019; McInerny et al., 2014), the need to actively build trust and transparency, the importance and value of face-to-face meetings, and building on pre-existing relationships were all mentioned, and all required or benefitted from more time availability (Carter et al., 2019; Lemos et al., 2012). For example, there were differences in the way relationships were built between the two projects, and different adaptive strategies to cope with challenges. The landslide project team members had not worked together previously and relationships evolved over multiple years, strengthened in particular by multiple, high-intensity, in-person workshops that often took place over days or weeks. In contrast, the cyclone bulletins did not have the luxury of time to develop such relationships, relying on the pre-existing close working relationship between two producer teams and rapidly developing relationships with the remote third producer team by nature of the intense development process and high-level of communication needed to produce the bulletins daily.

In both cases, the relationships between people were clearly important to the development of the project – not only between producers and users, but also within the producer team. These relationships were built or facilitated by physical proximity, previous relationships, openness, transparency, and time (Carter et al., 2019; Lemos et al., 2012). It should also be noted that whilst formal spaces such as workshops and meetings were important to develop these relationships, the informal "between-times" such as dinners, field trips, and out-of-office relaxation time were equally vital in developing stronger relationships.

### 6.6 Dealing with time pressure

Whilst the time available to develop the bulletins varied enormously between studies, the pressure of time was present in both. Notably, the challenges faced by producers of the bulletin, for example balancing the science and the needs of the user, did not get significantly easier with more time. The landslide project's overall scope to design, develop, test and operationalise a prototype system in one project window resulted in significant tension within the project team. Allowing time and funding for phases of new science development and testing and subsequent funding for the refinement and operationalisation of that science would reduce this tension and strengthen the impact of applied research projects.

Challenges related to time pressures remained a difficulty and a process that was worked through, no matter the time available. However, more time provided the opportunity for: producers to engage with users and the development of trust and shared understanding with users; the ability to test out and try new things; nuanced tailoring of products including careful consideration of content, language, and visualisation; and the development of preparatory materials to streamline and carefully think out processes in advance.

In terms of time pressures with issuing bulletins on a regular basis, interviewees recognised the benefits of prioritising preparedness activities using the time before the crisis or monsoon to develop protocols, templates, and deliver training to more than one person to ensure a smoother process (Patt and Gwata, 2002).

## 7 Conclusion

The two case studies provide evidence and insight into the development and production of forecast bulletins for institutional decision-makers. Key challenges from the case studies included: meeting user needs supported by strong science; communicating complex information (including uncertainty) clearly and effectively; and the limited time during crises to make changes and respond to feedback.

The solutions and approaches shared by the case study teams that can help address these challenges include:

- engage with users to understand what they need to know and work with them transparently;
- build an interdisciplinary team including social scientists, physical scientists, and practitioners or intermediaries;
- facilitate and build strong interdisciplinary collaboration, with good communication skills and an ability to work across disciplines;
- where possible, utilise time outside of intensive hazard or crisis periods to develop plans and protocols to improve efficiency of operational mechanisms;
- use interdisciplinary skills and delegation of roles to your advantage;
- be realistic and transparent within the project team and with external stakeholders about what can be achieved in the time available;
- value and actively build relationships between people in the team; and
- embrace an iterative approach by actively seeking feedback to optimise and improve the bulletins and processes.

Whilst the wider literature emphasises the importance of collaboration between disciplines, tailoring to users, and the importance of trust and protocols, there has been little operational guidance of *how* to do this in practice. Guidance is needed to provide structures and approaches for producers of forecasts to do this well, drawing on operational experience as well as academic published research. Guidance is also needed to define the specific skills needed from social science or intermediaries, better articulation of their benefits, and guidance on specific areas where the natural hazard community need to improve their skills. The findings indicate that whilst more research is needed, there is a significant body of experiential and intuitive knowledge and learning that already exists. Capturing this knowledge would be of significant benefit and interest to those developing forecast information.

## 8 Data availability

Data are available upon request to the lead author.

## 9 Author contributions

MB: conceptualisation, methodology, data collection, analysis, writing; AS: data collection, data analysis, writing; IN: literature review, data collection, transcribing; SB: data collection, writing, review and edit of paper; AD & LS: guidance on methodology and analysis, review of paper.

## 10 Competing interests

N/A

## 11 Acknowledgements

We would like to thank FCDO and NERC for funding provided as part of the SHEAR programme to conduct this study, and their funding that supported the development of bulletins via the LANDSLIP and FATHUM projects. We would also like to thank all those who took part in the interviews and participated in the study for their time and contributions.

We would particularly like to recognise and thank the following projects and organisations for their roles as producers and intermediaries and efforts in developing the bulletins and their collection and provision of resources to be used as secondary data analysis for this paper: LANDSLIP, FATHUM, Geological Survey of India, University of Bristol, European Centre for Medium-Range Weather Forecasts, and FCDO.

We would also like the following for reviewing the paper and providing substantial feedback: Emma Bee (British Geological Survey), Elisabeth Stephens (University of Reading), and Alessandro Mondini (Consiglio Nazionale delle Ricerche).

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
