# Peer review of "Development of forecast information for institutional decision-makers: landslides in India and cyclones in Mozambique"

_Geoscience Communication, 2021_

## Referee Comment (RC1)

**Requested review:** gc-2021-31.pdf

**Development of forecast information for institutional decision-makers: landslides in India and cyclones in Mozambique**

Based on two case studies, this paper identifies key learning in translating scientific forecasts into useful information focused on the process of developing forecast bulletins for decision-making.

The paper documents important experiential learning, primarily from the perspective of 'producers', underscoring the need for what have been increasingly recognised as key component of co-producing relevant risk information. While the findings may not be particularly novel for those who have taken part in similar resilience-building research consortia, the authors rightly highlight the need for greater practical guidance on how to develop user-relevant bulletins. Particularly valuable are reflections on the differences between producers and users with regard to content and inclusion of impact and advisory information within forecast bulletins, as well as issues surrounding piloting of risk information amongst resource-constrained at-risk populations.

**Abstract**

It could be extremely useful for the discussion to further highlight in the abstract the paper's important learning with regard to piloting of new risk information in resource-constrained environments.

**References**

References to relevant existing literature and resources could be strengthened to ensure the discussion builds on rather than repeats emerging learning on risk communication.

**General comments:**

It would be preferable to use the term 'user' rather than 'end user' and 'product' rather than 'end product'. It is increasingly recognised that development of relevant risk information requires ongoing dialogue and exchange of knowledge between 'producer' and 'user'. Rather than being seen as recipients of a finalised 'end of value chain' service, the active role of users in the ongoing process recognises this two-way process.

Section 2.1 and 2.2: These sections could be usefully reversed. Development or co-development coming before content, with content dependent on the specific user and decision-making context.

Section 5: Results

Given the extensive methodology employed, it could be useful to strengthen the results sections with key supporting quotes or testimony and, if feasible, some boxes of cross case study comparison summarising key similarities and differences.

**Comments by section and/or line**

Section 2.1: It would be good to recognise the context and cultural specificity of presentation and visualisation preferences.

Recognising cognitive challenges in communicating climate information: while recognising the differences in uncertainties in weather and climate information over timeframes, it may be useful here to reference work of Harold et al. (2019) Approaches to communicating climatic uncertainties with decision-makers; Harold et al. (2017). Enhancing the accessibility of climate change data visuals: Recommendations to the IPCC and guidance for researchers.

Section 2.2: Development section.

Dependent on what is meant by guidance, there are relevant resources intended to provide overall framing of weather and climate services that are not referenced, for example Carter et al. Manual: Coproduction of African weather and climate services. Likewise there is key additional literature that the discussion builds on, including for example, Patt and Gwata, 2002, Effective seasonal climate forecast application; Lemos et al, 2012, Narrowing the climate information gap.

The background literature section does not address, or insufficiently refers to, other key factors in the process of co-developing relevant risk information, including bulletins. As noted in the literature identified, these encompass, amongst others: equity (Vincent et al, Nature Climate Change 2020), timeliness, perceived and evaluated skill of risk information, inclusive communication reach and resources to act on the risk information provided (Patt and Gwata, 2002; Carter et al, 2019).

Section 3: Background

Line 188 and further through the paper: Challenges in piloting new risk products, initiated in research focused projects and which do not always encompass the safety net mechanisms for acting on trialled products. Given the increasing focus on action research, this is a key issue and would be good to further highlight.

Line 206-7: Be good to clarify: 'An analysis of the usefulness and use of the bulletins users is beyond the scope of this project': when the paper does refer to user feedback, for example lines 294-99. Without user feedback on usefulness, the paper would find it difficult to identify 'best' practice.

Section 5.2.4

The content here is extremely valuable, highlighting discussion on bulletin content and differences between producers and users.

Line 474: Critical skills: be good to include contextual knowledge, as noted in line 486.

Section 5.5 does not mention language challenges within the co-production process across countries.

Section 5.6: Interesting that no mention was made by key informants recognising the need to strengthen users' understanding of the extent and limits of scientific capacities, and to ensure this is included as a component of risk communication work (while noting reference to this is included in lines 602 and 631). Likewise no mention of the potential for participatory

evaluation of risk information, including user feedback on observations to inform model development.

Line 582: Training is mentioned in regard to 'sustainability', but the article includes limited discussion on integrating the required technical capacities within national institutions to ensure continuation of project-initiated services.

Section 6.1: extremely valuable reflection.

Line 671: need to recognise there are differences of opinion as to how the intermediary functions may best be sustained, i.e. rather than relying on an external agency, developing core intermediary functions within key 'producer' or 'user' institutions.

Section 6.3: Meeting user needs, the challenges in balancing scientific constraints and user needs is not new. Likewise the need for transparent communication of scientific confidence and certainties is a key principle in the Carter et al, Coproduction manual.

---

## Author Comment (AC1)

**RESPONSES TO REVIEWER #1**

*Based on two case studies, this paper identifies key learning in translating scientific forecasts into useful information focused on the process of developing forecast bulletins for decision-making.*

*The paper documents important experiential learning, primarily from the perspective of 'producers', underscoring the need for what have been increasingly recognised as key component of co-producing relevant risk information. While the findings may not be particularly novel for those who have taken part in similar resilience-building research consortia, the authors rightly highlight the need for greater practical guidance on how to develop user-relevant bulletins. Particularly valuable are reflections on the differences between producers and users with regard to content and inclusion of impact and advisory information within forecast bulletins, as well as issues surrounding piloting of risk information amongst resource-constrained at-risk populations.*

*Abstract*

1. *It could be extremely useful for the discussion to further highlight in the abstract the paper's important learning with regard to piloting of new risk information in resource-constrained environments.*

   The following sentence has been edited to the following text in the abstract at line 22:

   > "A major challenge was the difficulty of balancing science capabilities, including issues related to data scarcity, with user needs, which did not become significantly easier to deal with given more time availability."

   The following sentence has been added in the abstract at line 23:

   > "In particular, there were tensions between developing new forecast products that were urgently needed by users, against the limited time for testing and refinement of those forecasts, and the risk of misinforming decisions due to uncertainty of the information based on limited data."

*References*

2. *References to relevant existing literature and resources could be strengthened to ensure the discussion builds on rather than repeats emerging learning on risk communication.*

   The discussion section has thirteen existing references within the text that directly relate the discussion points from the study to published knowledge.

   The authors have expanded this to include eighteen additional references from the original bibliography as well as the added references outlined below in the author's responses from the publications suggested by the reviewer.

*General comments:*

3. *It would be preferable to use the term 'user' rather than 'end user' and 'product' rather than 'end product'. It is increasingly recognised that development of relevant risk information requires ongoing dialogue and exchange of knowledge between 'producer' and 'user'. Rather than being*

*seen as recipients of a finalised 'end of value chain' service, the active role of users in the ongoing process recognises this two-way process.*

> Noted and agree. All references to "end-user" have been changed to "user". All references to "end-product" have been changes to "product". Except in situations when directly quoting a citation or interviewee.

4. *Section 2.1 and 2.2: These sections could be usefully reversed. Development or codevelopment coming before content, with content dependent on the specific user and decision-making context.*

Agreed. These sections have been switched and the introduction to the literature review edited  to reflect this change in order.

**Section 5: Results:**

5. *Given the extensive methodology employed, it could be useful to strengthen the results sections with key supporting quotes or testimony and, if feasible, some boxes of cross case study comparison summarising key similarities and differences.*

Quotes from the interview have been added to the results section to demonstrate the points.

A table of comparison has been added to the paper. See reviewer #2 responses for details of the table.

**Comments by section and/or line**

6. *Section 2.1: It would be good to recognise the context and cultural specificity of presentation and visualisation preferences.*

> The following sentence has been added at line 98:

> > "Harold et al. (2017) recommend the following process for developing effective visuals: identify the main message, assess users' prior knowledge and thought process, choose visual formats familiar to users, reduce complexity where possible, build up information to provide visual structure, integrate text, avoid jargon, use cognitive design principles, and test visuals to check comprehension with users. Preferences for visual formats varies by users and by context, often influenced by factors including culture and educational or training background (Harold et al., 2017; Fleming et al., 2005)."

> The following citation has been added to the reference list:

> > Fleming, G., Gill, J., Muchemi, S., Al-Harthy, A.H.M., Cordeneanu, E., Diop, A.A., Martin, C., Lai, E., Groth, J., Palmer, S., and Cegnar, T.: Guidelines on weather broadcasting and the use of radio for the delivery of weather information, World Meteorological Organisation, 2005.

7. *Recognising cognitive challenges in communicating climate information: while recognising the differences in uncertainties in weather and climate information over timeframes, it may be useful here to reference work of Harold et al. (2019) Approaches to communicating climatic uncertainties*

*with decision-makers; Harold et al. (2017). Enhancing the accessibility of climate change data visuals: Recommendations to the IPCC and guidance for researchers.*

The authors would like to thank the reviewer for these publication suggestions. The authors have read the suggested papers and have added in reference to Harold et al. (2019) within the literature review section in five places to support statements related to uncertainty, use of visual information, trust and co-development. The authors have also edited the following sentence at line 86 to include reference to improved understanding as detailed in the paper:

> "The format and presentation of information critically influences the extent to which that information is understood. The use of images increases user risk perception (Bica et al., 2019; Anderson-Berry et al., 2018; Gough, 2017), understanding (Harold et al., 2019), and risk aversion (Visschers et al., 2009)."

Reference to Harold et al. (2017) has been added into the literature review section in five places to support statements related to visual imagery. The following sentence has been added at line 98:

> "Harold et al. (2017) recommend the following process for developing effective visuals: identify the main message, assess users' prior knowledge and thought process, choose visual formats familiar to users, reduce complexity where possible, build up information to provide visual structure, integrate text, avoid jargon, use cognitive design principles, and test visuals to check comprehension with users."

The following citations have been added to the reference list:

> Harold, J., Coventry, K., Visman, E., Diop, I.S., Kavonic, J., Lorenzoni, I., Jack, C., and Warnaars, T.: Approaches to communicating climatic uncertainties with decision-makers, Future Climate for Africa Guide, 2019.

> Harold, J., Lorenzoni, I., Coventry, K. R., and Minns, A.: Enhancing the accessibility of climate change data visuals: Recommendations to the IPCC and guidance for researchers, Tyndall Centre for Climate Change Research, Norwich, UK, 2017.

**Section 2.2: Development section.**

8. *Dependent on what is meant by guidance, there are relevant resources intended to provide overall framing of weather and climate services that are not referenced, for example Carter et al. Manual: Coproduction of African weather and climate services. Likewise there is key additional literature that the discussion builds on, including for example, Patt and Gwata, 2002, Effective seasonal climate forecast application; Lemos et al, 2012, Narrowing the climate information gap.*

The resources suggested by the reviewer are appreciated and support the literature review content related to the need for co-development and co-production. Whilst the suggested resources are incredibly useful guides to general principles related to co-production and climate services, they are not specifically addressing forecast bulletins as a product, therefore the authors are confident to continue to state within the paper that there is a lack of resources that

summarise or provide guidance specifically on the processes of setting up and developing natural hazard-related forecast bulletins for institutional decision makers.

The Carter et al. (2019) manual has useful guidance on how to co-develop weather and climate services that directly support the paper's findings and recommendations for developing forecast bulletins. Reference to Carter et al. (2019) has been added to the literature review section and discussion section to support statements on co-development. The following sentence has also been added at line 147:

> "Carter et al. (2019) outline a series of building blocks for co-production of weather and climate services including identifying key actors and building relationships, building common ground, co-exploring needs, co-developing solutions, co-delivering solutions, and evaluation."

The following sentence at line 681 has been edited to the following:

> "A recurring and evolving theme throughout both studies was the balance between science and user needs, reinforcing established principles for effective co-production (Carter et al., 2019)."

Reference to Patt and Gwata (2002) has been added to the literature review section and discussion section to support statements related to: credibility (including risks related to unskilful forecasts and issues around trust); cognition (understanding of forecast information); and procedures.

Reference to Lemos et al. (2012) has been added to the literature review section and discussion section to support statements related to: usability, saliency, long-term relationships, collaboration, two-way production, and iterative approaches.

The following citations have been added to the reference list:

> Carter, S., Steynor, A., Vincent, K., Visman, E., and Waagsaether, K.: Co-production of African weather and climate services, second edition, manual, Cape Town: Future Climate for Africa and Weather and Climate Information Services for Africa (https://futureclimateafrica.org/coproduction-manual), 2019.

> Patt, A., and Gwata, C.: Effective seasonal climate forecast applications: examining constraints for subsistence farmers in Zimbabwe, Global Environmental Change, 12, 185-195, 2002.

> Lemos, M.C., Kirchhoff, C.J., and Ramprasad, V.: Narrowing the climate information usability gap, Nature: Climate Change, review article, 001: lO.103B/NCLIMATE1614, 2012.

9. *The background literature section does not address, or insufficiently refers to, other key factors in the process of co-developing relevant risk information, including bulletins. As noted in the literature identified, these encompass, amongst others: equity (Vincent et al, Nature Climate Change 2020), timeliness, perceived and evaluated skill of risk information, inclusive communication reach and resources to act on the risk information provided (Patt and Gwata, 2002; Carter et al, 2019).*

The following sentences have been edited to include reference to equitable partnerships as discussed in Vincent et al. (2020) in the literature review section:

> Line 131: "Identification of bulletin users and equitable co-development with or tailoring to that audience is an integral part of an effective forecast product (Harrowsmith et al., 2020; Harold et al., 2017; Carter et al., 2019; Lemos et al., 2012; Vincent et al., 2020)."

> Line 144: "Robbins et al. (2019) explains that in order for forecasts to elicit the intended response, there needs to be regular "collaborative dialogue platforms" which require proper funding and operating procedures to be successful, as well as mechanisms to support equitable partnerships (Lemos et al., 2012; Vincent et al., 2020)."

> Line 146: "An effective forecast product requires long-term equitable partnerships between scientists, users/decision-makers and practitioners (Morss et al., 2005; Harold et al., 2019; Lemos et al., 2012; Vincent et al., 2020)."

The following sentence has been edited to include reference to timeliness:

> Line 144: "Robbins et al. (2019) explains that in order for forecasts to elicit the intended response, there needs to be regular "collaborative dialogue platforms" which require sufficient time to build trust and partnerships, proper funding and operating procedures to be successful, as well as mechanisms to support equitable partnerships (Lemos et al., 2012; Vincent et al., 2020; Carter et al., 2019)."

The following sentence has been edited to include reference to skill, or to support statements with new resources recommended by referee:

> Line 176: "Trust in the scientific forecasts themselves in terms of accuracy of predictions is also vitally important; evaluating, understanding, and communicating forecast skill transparently can support this (Harrowsmith et al., 2020, Patt and Gwata, 2002; Carter et al., 2019)."

> Line 634: "However, many of the physical scientists within the team see this sharing of untested knowledge as risky, particularly before validation of the forecast skill has been thoroughly conducted, as there is a risk of users making decisions based on untested science (despite being instructed not to), which could have severe and long-term consequences, as evidenced in real-life case studies and published literature (Patt and Gwata, 2002)."

The authors perceive that "inclusive communication reach" is now covered within the co-production content of the paper, particularly with the responses to reviewer #1's comments overall. Further exploration into this topic in depth will expand beyond the scope of the paper and unnecessarily lengthen the literature review as the paper is focused on institutional decision makers. The authors feel topics of public and community-level inclusive communication strategies and first/end-mile early warning is outside the focus of the paper on institutional decision makers.

The authors feel like the issue of resources to act on the information provided is beyond the scope of this paper to deal with in depth, particularly considering its updated length as a result of responses to reviewer's comments. The authors feel this topic is mentioned sufficiently

within the paper during the sections that relate to actionable information. These include the following parts:

> Line 134: "A multitude of factors make the development and communication of understandable and actionable forecast information incredibly complex, with complexity in the hazards themselves, alongside complex social, political and economic contexts (Patt and Gwata, 2002). Production of actionable forecasts necessitates understanding of the contexts in which this information is being shared and used (Harrowsmith et al., 2020)."

> Line 142: "McInerny et al. (2014) stress the importance of conducting targeted user research early on in the process to ensure products are relevant, understandable and actionable. Robbins et al. (2019) explains that in order for forecasts to elicit the intended response, there needs to be regular "collaborative dialogue platforms" which require sufficient time to build trust and partnerships, proper funding and operating procedures to be successful, as well as mechanisms to support equitable partnerships (Lemos et al., 2012; Vincent et al., 2020; Carter et al., 2019)."

> Line 156: "Wachinger et al. (2013) found that when communities are involved in designing and testing emergency plans, they are more motivated to listen and take action on information provided during a real event."

The following citation has been added to the reference list:

> Vincent, K., Carter, S., Steynor, A., Visman, E., and Wagsaether, K.L.: Addressing power imbalances in co-production, Nature: Climate Change, Comment, https://doi.org/10.1038/s41558-020-00910-w, 2020.

**Section 3: Background**

10. *Line 188 and further through the paper: Challenges in piloting new risk products, initiated in research focused projects and which do not always encompass the safety net mechanisms for acting on trialled products. Given the increasing focus on action research, this is a key issue and would be good to further highlight.*

    The focus of the paper is on the development of forecast information rather than the piloting of them. The following text has been added to section 3.0

    > "This paper also does not aim to explore the piloting and operationalisation of new risk products, and does not review practical and ethical issues of trialling new risk products amongst at-risk populations. This is noted as a limitation, and an area for further research."

11. *Line 206-7: Be good to clarify: 'An analysis of the usefulness and use of the bulletins users is beyond the scope of this project': when the paper does refer to user feedback, for example lines 294-99. Without user feedback on usefulness, the paper would find it difficult to identify 'best' practice.*

The sentence has been edited to the following:

"An analysis of the use of the bulletins by users is beyond the scope of this project (as users were not interviewed and the landslide project was operating as an experimental prototype system, with instructions given to users not to actively use the forecast information for decisions), but would be a valuable addition to the global body of knowledge on effective practice in this field."

**Section 5.2.4**

***The content here is extremely valuable, highlighting discussion on bulletin content and differences between producers and users.***

12. ***Line 474: Critical skills: be good to include contextual knowledge, as noted in line 486.***

Contextual knowledge added as a critical skill in line 474.

13. ***Section 5.5 does not mention language challenges within the co-production process across countries.***

> The following sentence has been added to line 506:

> > "It is worth noting that in these case studies, the bulletins were produced in English, with stakeholders who were all accustomed to using English as a working language. In other contexts, linguistic barriers will need to be considered as a potential barrier to effective collaboration."

14. ***Section 5.6: Interesting that no mention was made by key informants recognising the need to strengthen users' understanding of the extent and limits of scientific capacities, and to ensure this is included as a component of risk communication work (while noting reference to this is included in lines 602 and 631). Likewise no mention of the potential for participatory evaluation of risk information, including user feedback on observations to inform model development.***

> This did not come through strongly in the key informant interviews, but the paper does speak to these issues in a few key places. These are outlined below. Some of the following text has been edited to make clearer how users were mentioned within activities and feedback loops.

> > Line 297: "Feedback loops between producers, intermediaries and users shaped understanding of what information was understandable, relevant and useful for informing user action. Discussions centred on the bulletin development also helped to shape and inform users' understanding of the scientific capacities of the forecasts themselves."

> > Line 329: "Interviewees found that the process of seeking out and incorporating in-person user feedback was useful in strengthening relationships between users and producers, informing users of the science behind the forecasts, as well as for improving the usefulness and comprehension of the bulletin for users."

> > Line 529: "Both projects experienced challenges in the gap between what scale, detail and accuracy the users wanted to be able to make better decisions, and what the available data, science, and forecast technology (and project scope) was able to provide."

Line 535: "Both projects reflected on challenges in needing to manage user expectations regarding the level of detail and certainty that is possible to provide."

Line 546: "Both the cyclone and landslide teams discussed in interviews the issue of validation, highlighting the importance of verifying the performance of the model against the actual events in order to determine how accurately events were forecasted, and working with users to evaluate how useful the bulletin was in supporting effective humanitarian decision-making and response."

Line 630: "Social scientists within the team highlighted that involvement of users from the beginning could enhance the users' understanding of the limitations of the models and data and their inherent uncertainties."

15. **Line 582: Training is mentioned in regard to 'sustainability', but the article includes limited discussion on integrating the required technical capacities within national institutions to ensure continuation of project-initiated services.**

The following text has been added to this section:

"The two case studies adopted different approaches to sustainability. The bulletins for Cyclones Idai and Kenneth were developed to provide additional bespoke forecasts for those specific events, without a focus on embedding them within national institutions or sustaining them into the future. The bulletins for the landslide project were always intended to be sustained long-term, with the bulletin co-developed with the key national institutions with responsibilities for longer term application."

**Section 6.1:**

***Extremely valuable reflection.***

16. **Line 671: need to recognise there are differences of opinion as to how the intermediary functions may best be sustained, i.e. rather than relying on an external agency, developing core intermediary functions within key 'producer' or 'user' institutions.**

The following text has been added to section 6.2, line 674:

"There will be different perspectives on where intermediary roles should sit, with examples in this research of intermediaries being external to a given project, or embedded within a project team. Regardless of location, it is important to ensure these roles and skillsets are emphasised, especially in institutions where such applied or social science roles may not be currently prioritised."

17. **Section 6.3: Meeting user needs, the challenges in balancing scientific constraints and user needs is not new. Likewise the need for transparent communication of scientific confidence and certainties is a key principle in the Carter et al, Coproduction manual.**

The following sentence at line 681 has been edited to the following:

"A recurring and evolving theme throughout both studies was the balance between science and user needs, reinforcing established principles for effective co-production (Carter et al., 2019)."

---

## Author Comment (AC2)

**RESPONSES TO REFEREE #2**

*The paper highlights and responds to a gap in research on the process of developing forecast bulletins for decision-makers i.e. professional, institutional users. It includes many practical and helpful insights from the process, as examined in two different cases.*

*Some comments and suggestions are given below:*

1. *Context: Consider using diagrams/figures to concisely convey the various flows of information and stakeholder relationships (Users, Intermediaries and Producers) for clarity.*

    We have added the following diagram to the paper with the accompanying citation to improve clarity. It has been added after line 64 where the paper outlines producer, intermediary and user roles. It has been cited within the paper at line 64 (Figure 1).

[Figure]

    **"Figure 1 Diagram indicating common relationship patterns between the roles of producers, intermediaries and users in bulletin development and production. Arrows indicate typical flows of information."**

    All figure numbering has been updated to include this figure.

2. *Methodology: The choice of cases appears to be fairly pragmatic and the methodology would benefit from further explanation and justification of the choice and the implications for the subsequent analysis.*

    We have added the following text to the Data and Methodology section at line 270.

    "The choice of the case studies was based on the authors involvement within the SHEAR programme. The authors of this paper have occupied various roles within the SHEAR programme including: consortium members in the LANDSLIP and FATHUM projects; team members involved in the development of the bulletins; and/or acted as Knowledge Brokers of the SHEAR programme. In the process of carrying out these roles, the authors witnessed challenges, and commonalities and differences between approaches and solutions for each case study and identified these examples as presenting an opportunity for learning about the process of developing bulletins from those who were involved."

The following paragraph in the submitted paper goes onto outline the opportunities this provides and also outlines the efforts the author team have made to ensure the results are based on the data gathered and limit biases within the authorship team.

> "The authors of this paper bring a range of roles and a unique dual perspective to these case studies, bringing together core team members of both case studies (bringing an insider perspective), alongside those outside of the core projects who have engaged with those initiatives and teams over several years as Knowledge Brokers of the wider SHEAR programme (bringing a semi-outsider perspective). The authors have made efforts to focus reporting of the results directly from the data sources, ensuring all perspectives are represented, whilst also reflecting on useful learning during the discussion section, to bring in their unique position and experiential knowledge."

3. *Results: A strength is the level of detail and granularity provided in the results section. Giving specific examples, for example the choice of words or colours, transparently allows the reader to see into the process of co-production and development of bulletins and means that these insights are not lost in over-generalisation. However, I think that consistency and signposting of when you are writing about which case and the similarities and differences, could be improved for clarity. Consider whether summary tables could be helpful.*

The authors have checked through the paper and edited sentences so that it is clearly stated which case study is being described (using cyclone or landslide as the distinguishing characteristic for consistency).

We have also added in the following table in section 5.2 Bulletin content (line 333):

> "Table 3 summarises some of the key features and changes to the cyclone and landslide bulletins, which are described in more detail in the following sections.
>
> **Table 3 Key features and changes to the content of the bulletins, including layout, text, visuals, and information."**

| Content | Both | Landslide bulletin | Cyclone bulletin |
|---------|------|--------------------|--------------------|
| Layout | Summary information at the beginning. More detailed information provided later. | Evolved from 1 page to 2 pages. First page providing changing information, second page containing static information. | Cyclone Idai bulletins evolved from 9 pages to 13-15 pages, as new information added. Cyclone Kenneth bulletins evolved from 5 pages to 10 pages. Summary information as bullet points on first page. |

| | | | |
|---|---|---|---|
| | | | Update section added, summarising changes since last bulletin on first page. |
| Text | Simplification of terminology. Reduction in the amount of text provided. Text accompanying visuals to explain them. | Text descriptions of each day's forecast provided instead of levels of warnings. Changed title from "warning" to "forecast" and "experimental" added. Forecast level terminology changed from [Widespread (most places); Fairly widespread (many places); Scattered (a few places); and Isolated] to [Less likely; likely; more likely; most likely] then to [Very high; High; Moderate; Low]. Terminology explanations provided in key. | Summary first page layout edited to be easier to read. Methodology section removed (remained available as static information). |
| Visuals | Labelling of key places (particularly if mentioned in text) and administrative areas onto maps. Increase in the number of visuals (maps and graphs) with keys and supplementary text. | Removal of weather forecast maps and focus on landslide forecast maps. Forecast key colours changed to IMD traffic light colour system. 'Spots' of colour added to maps where warning level is higher/lower than assigned administrative level. Changed to freestyle shapes. Landslide susceptibility map and text included on second page. Changed to greyscale, then to red tones. | GLoFAS colour scheme changed to traffic light system. Map of area focusing on added to first page. Various maps and graphs added: flood hazard map; graph of temporal forecasts from ECMWF; probability of exceedance of severe flood level; timeline of observed flood extent maps Satellite imagery maps added then removed. Simplification of graphs and maps. |
| Information | Evolving content of type of information. No advice included. | Warning, vulnerability, impact and action content removed. | Evolving to include three main pieces of information: 1) meteorological forecast; 2) flood forecast; and 3) |

| | | Important information section added to second page with information on uncertainty and caveats/limitations. Added disclaimer in red text below title. Rivers and roads added to static maps. | flood hazard and population exposure information. |
|---|---|---|---|

**4. *Second paragraph of 5.3 Information vs. advice - seems to be more about complexity. Is this a theme in itself?***

The authors reflected that this second paragraph was a previously identified separate theme on complexity that was merged with the content on information and advice as it is a linked topic. We have separated these topics back out by inserting a separate sub-title for the second paragraph (5.4 Communicating complexity). Section numbering has been updated to reflect this addition.

This topic is also mentioned at multiple stages of the paper, so the authors suggest there is not a need to include anything in addition to the current dialogue. For example, complexity is discussed in the following sections: abstract (line 18), literature review (line 134), results (line 402), considerations (line 666) and conclusions (line 752):

"Both case studies experienced challenges dealing with uncertainty, complexity, and whether to include advice."

"A multitude of factors make the development and communication of understandable and actionable forecast information incredibly complex, with complexity in the hazards themselves, alongside complex social, political and economic contexts."

"Given the complexity of the information being provided in the graphics, and the range of possible interpretations of visual information, explanatory text was deemed essential by producers and intermediaries (and from user feedback) to enable end-users to understand the context and meaning of the maps and colours in the bulletin."

"Where this understanding and appreciation of added value was lacking to begin with (in the case of some physical scientists), it evolved over time as pressures to operationalise bulletins increased awareness of the importance and complexity of communicating useful information to users."

"Key challenges from the case studies included: meeting user needs supported by strong science; communicating complex information (including uncertainty) clearly and effectively; and the limited time during crises to make changes and respond to feedback."

5. *Considerations: This more discursive section aims to bring together insights from the results for others developing forecasts and bulletins for natural hazards. A topic of importance that I feel is not adequately addressed is that of accountability - who (among all stakeholders - producers, intermediaries and users) is ultimately accountable for the information provided and its impacts? And how has this been considered throughout the bulletin development process?*

This topic is partially covered in section "6.3 Meeting user needs" (line 690). The authors recognise accountability is an important topic, so suggest moving the paragraph from line 690 to a new section "6.4 Mandates and responsibilities" and editing the text to the following content. Section numbering has been updated to reflect this addition.

"There were tensions in both studies between balancing science and user needs not only because of what is possible for scientists to provide, but also influenced by tensions related to the mandate and purpose of science and scientists (specifically physical science forecasters), and also by the aims, scope, and restrictions of funded projects. In both studies, there were challenges related to users requesting information that was beyond the scope of the project, for example, the inclusion of exposure, impact, and vulnerability data or assessments which could be used to influence actions that affect people's lives.

The official responsibilities of producing forecast information were different for each project. The cyclone bulletins were produced by non-responsible institutions at the request of a key stakeholder. As such one of their main focus points was to ensure scientific rigour in the information they provided, to protect institutional reputation, but they were not officially responsible or mandated with providing the information - it was supplementary to formal mechanisms and information.

For the landslide bulletins, this was more complex as the project lifetime covered a period when the institution that would undertake production of bulletins beyond the project funding was undergoing a major shift in their institution's role and official mandate during the project lifetime, changing from their previous focus on response to landslides towards the provision of information in advance of landslides. This change in mandate required a significant institutional culture shift and a rapid learning curve to overcome the initial lack of experience, familiarity and confidence in issuing forecast information.

Landslide project interviews highlighted the impact of institutional mandates and responsibilities on the bulletin, emphasising that the producer's responsibility was to provide forecast information, and not to issue warnings. This directly affected the content of the bulletin: the terminology of "forecast" rather than "warning" was carefully chosen, it was decided not to provide (or update) vulnerability information in the bulletin, and it was decided not to provide advice on actions to be taken in response to warnings.

In published literature and real-world examples, there is a tension in not just what science can provide, but whether they should provide it at all. This comes to the fore particularly when science is used to make decisions alongside other evidence (Frick and Hegg., 2011). When these types of decisions are the role and responsibility of government officials, but need to be informed by science, then scientists need to be careful in considering what they provide, how they provide it, and how to communicate it (Kox et al., 2018). There needs to be a clear and transparent agreement and awareness of the difference in roles, responsibilities, and mandates of the producers of forecasters compared to that of the institutional decision-makers (Sukhwani et al., 2019). This is vital in developing and protecting forecast producer's scientific reputation and the users' trust in their abilities."